# Data integration across conditions improves turnover number estimates and metabolic predictions

Philipp Wendering [1,2,3], Marius Arend [1,2,3], Zahra Razaghi-Moghadam[2] & Zoran Nikoloski [1,2] ✉

Turnover numbers characterize a key property of enzymes, and their usage in constraint-based metabolic modeling is expected to increase the prediction accuracy of diverse cellular phenotypes. In vivo turnover numbers can be obtained by integrating reaction rate and enzyme abundance measurements from individual experiments. Yet, their contribution to improving predictions of condition-specific cellular phenotypes remains elusive. Here, we show that available in vitro and in vivo turnover numbers lead to poor prediction of condition-specific growth rates with protein-constrained models of *Escherichia coli* and *Saccharomyces cerevisiae*, particularly when protein abundances are considered. We demonstrate that correction of turnover numbers by simultaneous consideration of proteomics and physiological data leads to improved predictions of condition-specific growth rates. Moreover, the obtained estimates are more precise than corresponding in vitro turnover numbers. Therefore, our approach provides the means to correct turnover numbers and paves the way towards cataloguing kcatomes of other organisms.

Genome-scale metabolic models (GEMs) together with advances in constrained-based modeling have led to an improved understanding of how cellular resources are used to fulfill different cellular tasks[1-3]. Recent advances are largely propelled by the development of protein-constrained GEMs (pcGEMs) in which the catalytic capacities of individual enzymes are linked to the allocation of enzyme abundances[4]. Such models have led to more accurate predictions of maximum specific growth rates on different carbon sources[5-7], flux distributions[7], and other complex phenotypes[8] in *Escherichia coli* and *Saccharomyces cerevisiae*. However, the development of pcGEMs critically depends on the integration of organism-specific enzyme turnover numbers, $k_{cat}$, comprising the kcatome of an organism[9].

Measuring the kcatome of an organism based on in vitro characterization is limited due to the impossibility to purify specific enzymes, lack of availability of substrates, and knowledge of required cofactors, such that their relevance for studies of in vivo

phenotypes remains questionable[10,11]. Proxies for in vivo turnover numbers also termed maximal apparent catalytic rates, can be estimated by combining constraint-based approaches for flux prediction with measurements of protein abundance under different growth conditions or genetic modifications[12-14]. The results from this approach, which entails ranking condition-specific estimates that use individual data sets, have shown that the proxies for in vivo turnover numbers generally concur with in vitro $k_{cat}$ values in *E. coli*[12]. However, applications with data from *S. cerevisiae*[15] and *A. thaliana*[16] indicated that these proxies for in vivo turnover numbers do not reflect in vitro measurements. Another approach to estimate the kcatome relies exclusively on the machine and deep learning methods that use a variety of features of enzymes (e.g. network-based, structure-based, and biochemical)[17-19], resulting in predictive models that can explain up to 70% of the variance in turnover numbers obtained in vitro.

[1]Bioinformatics, Institute of Biochemistry and Biology, University of Potsdam, Potsdam, Germany. [2]Systems Biology and Mathematical Modelling, Max Planck Institute of Molecular Plant Physiology, Potsdam, Germany. [3]These authors contributed equally: Philipp Wendering, Marius Arend. ✉e-mail: nikoloski@mpimp-golm.mpg.de

The estimates of turnover numbers are integrated into metabolic models by different constraint-based approaches that have been grouped into coarse-grained (e.g. MOMENT[5], sMOMENT[20], eMOMENT[21], and GECKO[7], which all result in the same feasible space in protein limited growth scenarios) and fine-grained (e.g. resource balance analysis[1] and ME-models[2,3]). Of these, GECKO[7] has been adopted in several recent studies due to the elegantly structured formulation of the protein constraints. In addition, GECKO allows for the integration of protein contents and correction factors that account for the mass fraction of enzymes ($f$) included in the model as well as the average in vivo saturation ($\sigma$) of all enzymes, facilitating the development of condition-specific models. While data-driven estimation of in vivo turnover numbers improves the coverage of $k_{cat}$ values in pcGEMs, the available estimates usually lead to over-constrained models when using the allocation of total protein mass, not considered in flux balance analysis (FBA)[22,23].

Here, we propose PRESTO (for protein-abundance-based correction of turnover numbers), a scalable constraint-based approach to correct turnover numbers by matching predictions from pcGEMs with measurements of cellular phenotypes− simultaneously−over multiple conditions. As a constraint-based approach, PRESTO facilitates the investigation of the variability of the proposed corrections. We show that predictions of growth by pcGEMs of *S. cerevisiae* with turnover numbers corrected by PRESTO are more accurate than those based on the models that include $k_{cat}$ values corrected based on a contending heuristic that relies on enzyme control coefficients[22]. We also demonstrate that the same conclusions hold when enzyme abundances are integrated into the *E. coli* pcGEM using PRESTO. Therefore, PRESTO paves the way to broaden the applicability of pcGEMs for organisms with biotechnological applications and to arrive at genotype-specific estimates of the kcatome.

## Results
### Protein-abundance-based correction of turnover numbers
For a given data set of protein abundances over a set of conditions, the enzymes with turnover numbers in a pcGEM can be partitioned into three groups. For instance, a data set of protein abundances that was recently used to estimate in vivo turnover numbers in *S. cerevisiae*[15] includes 45%, 41%, and 14% measured overall, at least one (but not all),

and none of the 27 used conditions, respectively. Therefore, there is then different data support for correcting the $k_{cat}$ values of these classes of proteins. PRESTO relies on solving a linear program that minimizes a weighted linear combination of the average relative error for predicted specific growth rates and the correction of the initial turnover numbers integrated into the pcGEM (Fig. 1, see the "Methods" section). It further employs $K$-fold cross-validation (here, $K = 3$) with 10 repetitions while ensuring a steady state and integrating protein constraints for proteins measured overall conditions (Fig. 1, see the "Methods" section). The training set of conditions is used to generate a single set of corrected in vitro $k_{cat}$ values, by using the respective in vivo protein abundances. The resulting corrected $k_{cat}$ values are in turn used to determine the relative error of the predicted specific growth rate for each condition in the test set using flux balance analysis with the pcGEM, while only constraining the total protein content and measured uptake rates. The relative error of the predicted specific growth rate along with the sum of introduced corrections is lastly used to select the value for the tuning parameter $\lambda$ in the objective function of PRESTO, as done in machine learning approaches that rely on regularization.

### PRESTO outperforms a contending heuristic in *S. cerevisiae*
To determine the performance of PRESTO and compare it to that of contending heuristics, we used a data set comprising protein abundances and exchange fluxes from 27 diverse conditions, as supported by the principal component analysis (Supplementary Fig. 1). Application of PRESTO with a pcGEM of *S. cerevisiae* with initial in vitro turnover numbers obtained from BRENDA resulted in a mean relative error of 0.68 from the cross-validation procedure, yielding a correction of on average 213 turnover numbers (Supplementary Fig. 2a). For the *S. cerevisiae* pcGEM, we found a value of $10^{-7}$ for the parameter $\lambda$ in the PRESTO objective provides the optimal trade-off between both the relative error and the sum of introduced corrections (see the "Methods" section). Moreover, we observed a high overlap between the sets of proteins with corrected turnover numbers in the cross-validation (average Jaccard distance of 0.07 (Supplementary Fig. 2b, c)), suggesting that the integrated data from different conditions point to a specific subset of enzymes that need to be corrected to improve performance of growth prediction.

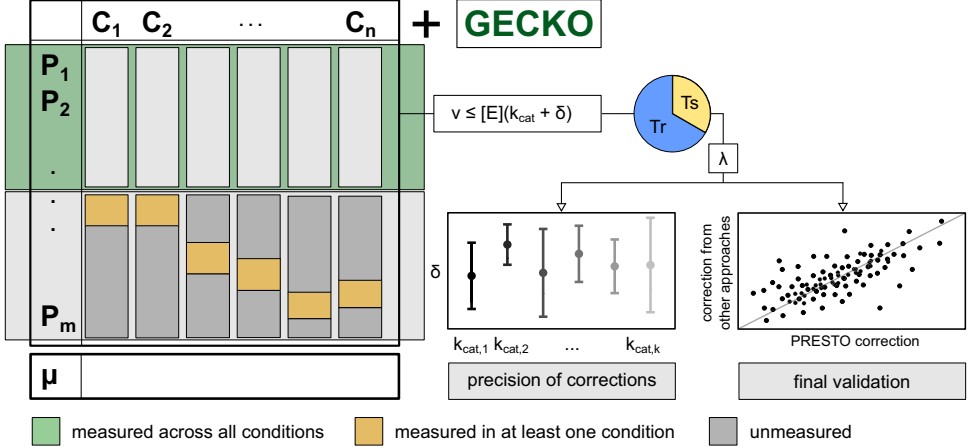

**Fig. 1 | Schematic overview of the PRESTO approach for $k_{cat}$ correction.** The approach uses a GECKO-formatted pcGEM[7] containing turnover numbers from BRENDA[48]. Using available data from $n$ experimental conditions, $n$ condition-specific models are generated using nutrient uptake rates and protein contents. PRESTO then uses data on abundances for the enzymes measured across the $n$ investigated conditions and solves a linear program that minimizes a weighted sum of two objectives, the relative error to measured specific growth rates and the sum of positive $k_{cat}$ corrections, $\delta$. The optimal weighting factor, $\lambda$, which modulates the trade-off between the two objectives, is then determined by cross-validation (Tr: training set; Ts: test set), choosing the parameter, which is associated with the lowest average relative error. Using the optimal value for $\lambda$, PRESTO combines all models for the experimental conditions to find a $k_{cat}$ correction for each enzyme with measured abundance. Last, the precision of $\delta$ values is assessed by variability analysis as well as by sampling and corrected $k_{cat}$ values are validated by comparing them to values obtained from other approaches.

Unlike PRESTO, GECKO implements a heuristic for the correction of turnover numbers that are based on the objective control coefficient calculated for each protein in a given condition (Supplementary Fig. 3)[22]. The control coefficient of a protein is determined by increasing the turnover number by 1000-fold and scoring the effect on the predicted specific growth rate. The proteins are then ranked in decreasing order of their control coefficients, and the turnover number of the first enzyme in the list is changed to the maximum value found in BRENDA for this enzyme across all organisms. This procedure is repeated with the remaining enzymes until the pcGEM predicts a growth rate that is at most 10% smaller than the measured specific growth rate for that condition or no additional $k_{cat}$ value that strongly constrains the solution can be found (Supplementary Fig. 3). This leads to condition-specific sets of corrected $k_{cat}$ with large intersections or full containment over a considered order of conditions (Supplementary Fig. 4a).

In contrast to this procedure, PRESTO corrects at once the turnover numbers of multiple enzymes that are measured in all investigated conditions by simultaneously leveraging the data from the different conditions, considerably reducing runtime and the number of solved problems. As a result, rather than deriving condition-specific corrected $k_{cat}$ values, which are difficult to use in making predictions for unseen scenarios or for building large-scale kinetic metabolic models[24,25], PRESTO results in a single set of corrected $k_{cat}$ values.

We compared the performance of PRESTO with the heuristic implemented in GECKO in three modeling scenarios that consider: (i) only condition-specific total protein content, (ii) both total protein content and uptake constraints, and (iii) additional constraints from abundances of enzymes measured in all conditions (Fig. 2). For corrections of turnover number from PRESTO, we observed that the relative error spans the range from 0.15 to 0.88 in the least constrained scenario (i) (Fig. 2a) and from 0.69 to 0.98 in the most constrained scenario (iii) (Fig. 2c). In contrast, the relative error with the corrections of turnover numbers from the GECKO heuristic is in the range from 0.96 to 1.00 in scenario (iii) (Fig. 2c). In addition, in scenario (iii), the median relative error in the case of the GECKO heuristic for each condition is larger than the relative error of the PRESTO predicted specific growth rate (Fig. 2c). We observed that predictions from FBA, considering enzyme abundances, without a constraint on the total protein content, led to an average relative error of 0.70 with $k_{cat}$ values corrected according to PRESTO and 0.99 with $k_{cat}$ values corrected according to GECKO (Supplementary Table 1).

We also performed a sensitivity analysis by investigating a smaller value, of $10^{-10}$, for the weighting factor $\lambda$ used in the PRESTO objective. We found that when the weighting factor is $10^{-10}$ (at which the total corrections of the initial $k_{cat}$ values plateaus), the relative errors from PRESTO cross-validation can be further decreased to 0.69 considering the constraint on the total protein content, with no effects on the other findings (Supplementary Fig. 2a). We also note that the relative error lies in the range from 0.35 to 0.80 over the considered weighting factors in the range from $10^{-14}$ to $10^{-1}$. Together, these results demonstrated that $k_{cat}$ values corrected according to PRESTO provide better model performance than the values obtained by the contending heuristic in the case of S. cerevisiae in the scenarios where all available data are integrated into the model constraints.

### PRESTO provides precise corrections of turnover numbers

In the following, we investigated the precision of the corrected $k_{cat}$ values from the application of PRESTO to data and a pcGEM model of S. cerevisiae. To this end, we determined the range that the correction of the $k_{cat}$ value of each enzyme can take while fixing the relative error in specific growth rate and total corrections from the optimum of PRESTO (see the "Methods" section). Moreover, we complemented this analysis by sampling corrected $k_{cat}$ values that achieve the

optimum of PRESTO with two values of the weighting factor $\lambda$ of $10^{-7}$ and $10^{-10}$.

In the case of the corrected $k_{cat}$ values for S. cerevisiae with a weighting factor of $10^{-7}$, we found that the $k_{cat}$ values with the largest corrections are more precisely determined (Supplementary Fig. 5). In addition, the sampled corrections per enzyme show an average Euclidean distance to the respective mean of $4.88 \text{ s}^{-1}$, indicating that the corrected values are more precise than the values in BRENDA, exhibiting an average Euclidean distance of $27.54 \text{ s}^{-1}$ to the mean per EC number (Supplementary Fig. 6). Importantly, while $k_{cat}$ values with smaller correction showed larger variability, the 25 and 75 percentiles of the sampled corrections for 42 enzymes are concentrated around those resulting from PRESTO. Repeating the analysis with a weighting factor of $10^{-10}$ showed that the larger total corrections of the initial $k_{cat}$ values resulted in also larger variability for the corrections over all $k_{cat}$ (Supplementary Fig. 7). Here, too, for 62 enzymes the 25 and 75 percentiles of the sampled corrections are concentrated around those resulting from PRESTO. Therefore, we concluded that the corrections from PRESTO are precise and can be used in downstream analyses.

### Pathways enrichment for corrected turnover numbers

In pcGEMs generated by the GECKO toolbox[7], turnover numbers are assigned to each of the enzymes in the GEM using a fuzzy matching algorithm. It takes into account the organism, substrate, and EC number of a BRENDA entry. When we investigated the magnitude of the turnover number correction dependent on the quality of the match between BRENDA entry and the corresponding enzyme, we found that $k_{cat}$ values measured in S. cerevisiae were associated with smaller corrections than those from other organisms (Supplementary Fig. 8a).

To check which metabolic processes are more likely to require correction of in vitro $k_{cat}$ values, we next conducted an enrichment analysis based on the KEGG pathway terms linked to corrected $k_{cat}$ values (see the "Methods" section). The most prominent pathway in this analysis was the synthesis of secondary metabolites, particularly the synthesis of cofactors and terpenoids (Fig. 3a). However, several terms linked to central carbon metabolisms, such as the tricarboxylic acid cycle and oxidative phosphorylation, were also significantly enriched. Interestingly, amino acid synthesis was the only term linked to nitrogen metabolism that came up in this analysis, although many pathways of nitrogen metabolism were among the tested terms. This analysis suggested that particularly in vitro turnover numbers in carbon metabolism need to be corrected, due to the underestimation of in vitro assays.

### Comparison of turnover number corrections from GECKO

Next, we aimed to identify the extent to which the corrected $k_{cat}$ values differ between PRESTO and the GECKO approach. To this end, we determined the intersection of enzymes with $k_{cat}$ values corrected manually[7], by PRESTO, and by the GECKO heuristic. For this comparison, we considered the union of all condition-specific corrected $k_{cat}$ values from the GECKO approach. With the weighing factor $\lambda = 10^{-7}$, PRESTO adapted the $k_{cat}$ values of 48% of enzymes corrected by the GECKO heuristic (Fig. 3b, Supplementary Data 1). We did not find a significant Spearman correlation ($\rho_S = 0.17$, $P = 0.45$) between the log-transformed $k_{cat}$ values in this intersection (Fig. 3c), owing to the different principles employed in the two procedures. To determine the pathways that comprise enzymes whose turnover number are corrected by GECKO and PRESTO, we next repeated the pathway enrichment analysis for the enzymes in the overlap. Among the significant terms, like in PRESTO, we again found 2-Oxocarboxylic acid, amino acid, and secondary metabolism to be enriched (Fig. 3a, S9). However, the more specific pathway terms were associated with pathways that are part of carbohydrate metabolism and aromatic amino acid metabolism corrected by both approaches (Supplementary Fig. 9,

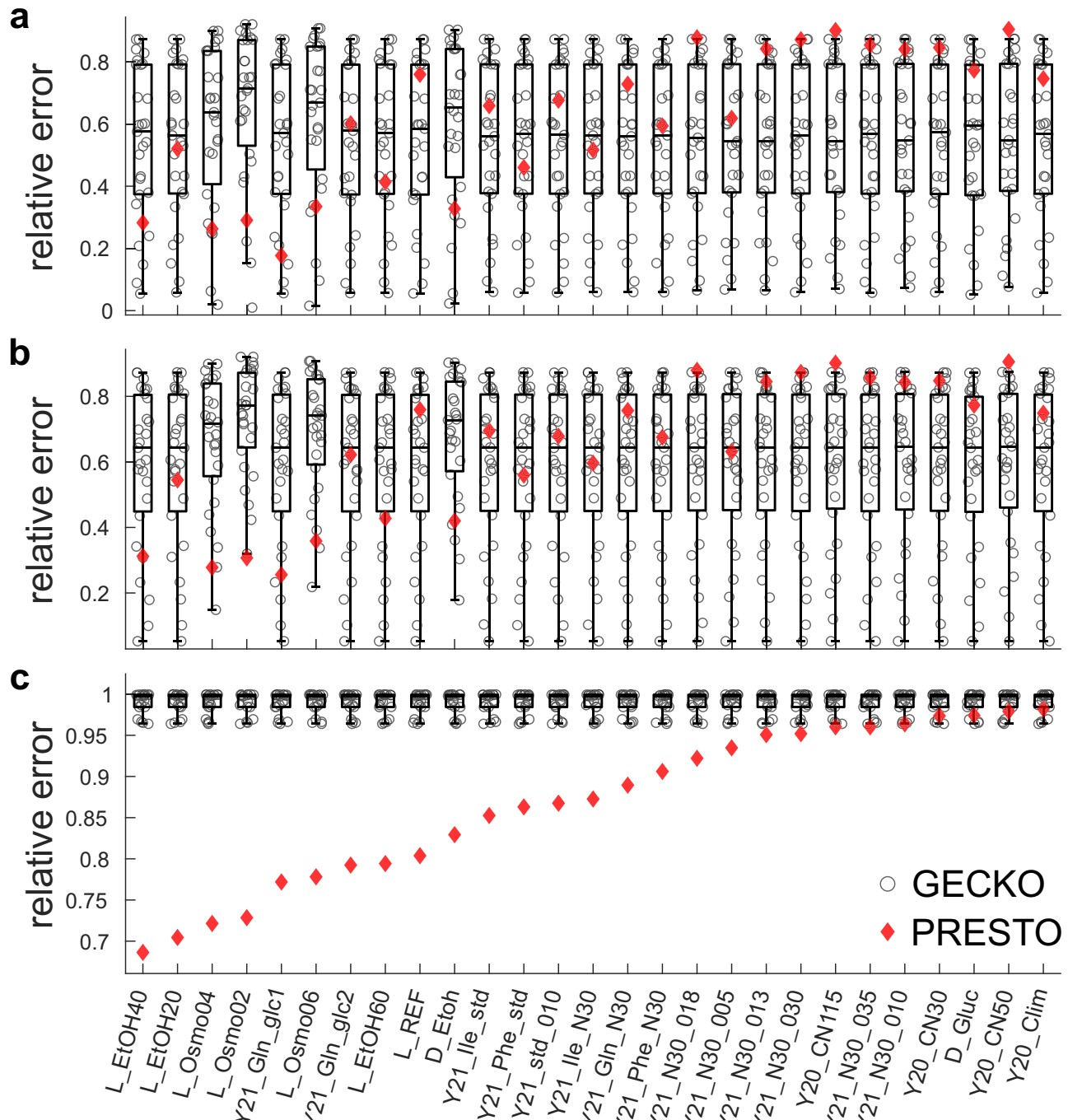

**Fig. 2 | Comparison of predicted growth of *S. cerevisiae* from pcGEMs with $k_{cat}$ corrections from GECKO and PRESTO.** Condition-specific pcGEMs with corrected $k_{cat}$ values generated by the GECKO heuristic were used to predict the specific growth rate for each condition ($n = 27$, **a**, **b**). The boxplots indicate the distribution of the relative error resulting from each set of condition-specific corrected $k_{cat}$ values obtained from the GECKO heuristic. Relative prediction error from each set is indicated by a circle. The red diamonds show the relative error of the predicted specific growth rate from the PRESTO model ($\lambda = 10^{-7}$) by using the single set of corrected $k_{cat}$ values in the respective pcGEM. **a** Only the measured total protein pool was used to constrain the solution and condition-specific uptake rates were bounded by 1000 $\frac{mmol}{h\,gDW}$; **b** measured uptake rates were also considered; **c** abundances of enzymes measured in all conditions were used as additional constraints. The compared pcGEMs in each condition ($n = 19$) used the same respective biomass reaction, GAM, $\sigma$, and $P_{tot}$ values (see the "Methods" section). L: Lahtvee et al.[34], D: Di Bartolomeo et al.[37], Y20: Yu et al.[35], Y21: Yu et al.[36]. Middle line and boxes in the box charts in panels **a**–**c** indicate the median and 25th and 75th percentiles, respectively. Outlier values (circles outside the whisker range) are more than 1.5× the interquartile range away from the top or bottom of the box, and whiskers connect the lower or upper quartiles with the non-outlier minimum or maximum. Source data are provided as a Source Data file.

Supplementary Data 2). In addition, the intersection between enzymes with manually corrected values and those corrected by the GECKO heuristic is higher than with PRESTO. This is expected since the manual curation is partly aimed at correcting the most constraining turnover numbers[7].

We also compared the $k_{cat}$ values adjusted by GECKO against estimates of in vivo $k_{cat}$ values obtained by parsimonious FBA (pFBA) using the same proteomics data[15] (Supplementary Fig. 10a, b). We confirmed the low correspondence ($\rho_S = 0.23$) between the $k_{cat}$ values obtained from BRENDA, included in the GECKO model without manual

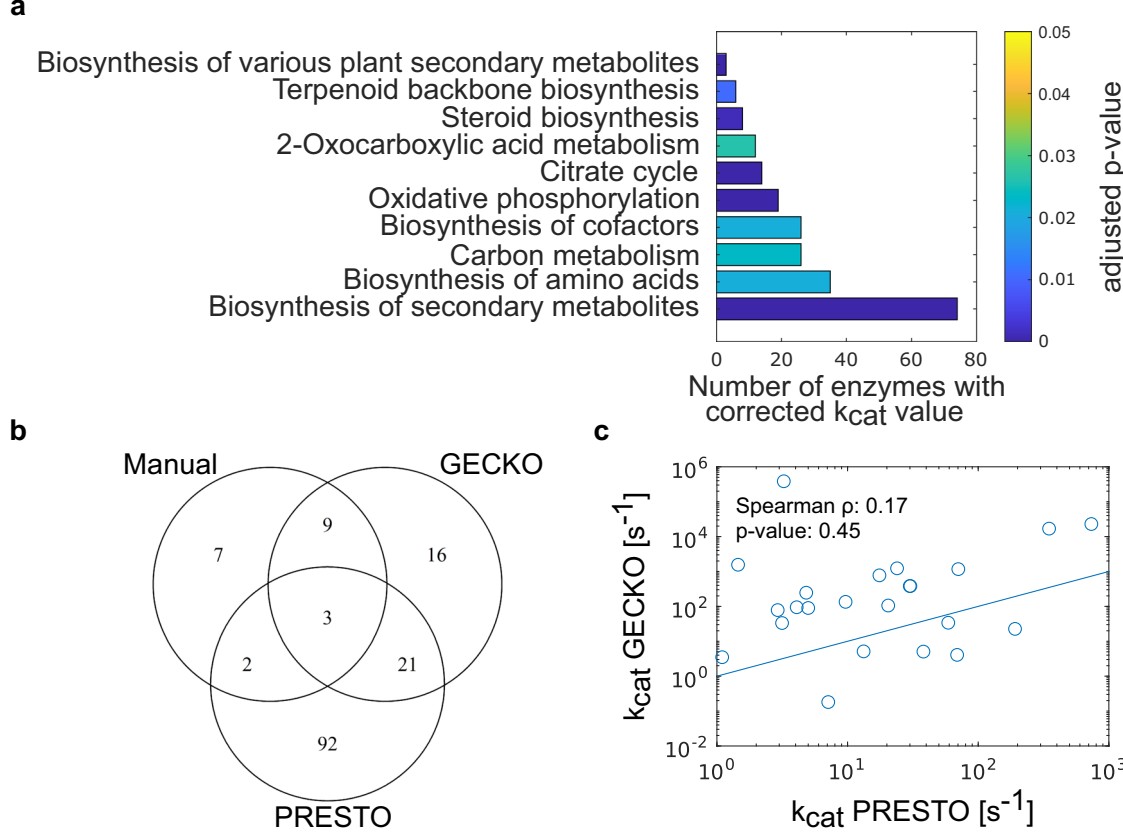

**Fig. 3 | Comparison of enzymes with corrected $k_{cat}$ values by both GECKO and PRESTO. a** KEGG Pathway terms significantly enriched in the set of enzymes corrected by PRESTO ($\lambda = 10^{-7}$) in the *S. cerevisiae* pcGEM. The *x*-axis gives the number of corrected enzymes linked to the given term. The one-sided *p*-values were calculated using the hypergeometric density distribution and corrected for multiple hypothesis testing using the Benjamini–Hochberg procedure[45]. **b** Venn diagram showing the overlap of enzymes whose $k_{cat}$ values were manually corrected[7] ("Manual"), automatically corrected by the GECKO heuristic in any of the conditions ("GECKO"), or corrected by PRESTO ("PRESTO"). **c** Log-transformed $k_{cat}$ values corrected using both the GECKO heuristic and PRESTO are not associated (Spearman correlation coefficient of 0.166, *p*-value = 0.45). Source data are provided as a Source Data file.

modifications, and the in vivo $k_{cat}$ estimates. As expected, the correspondence of the estimated in vivo $k_{cat}$ values to the turnover numbers corrected based on PRESTO was higher ($\rho_S = 0.34$). To investigate how these estimates perform as model parameters, we also generated a pcGEM in which BRENDA values were substituted by in vivo $k_{cat}$ values from pFBA[15], whenever available. In scenarios without enzyme abundance values, this model performed worse than that including the $k_{cat}$ values corrected by PRESTO as well as the model combining the maximum of all condition-specific GECKO corrections (Supplementary Fig. 11a, b). In the enzyme abundance-constrained scenario, the model with in vivo turnover numbers estimated by pFBA performed slightly better than GECKO but still only achieved a minimum relative error of 0.93, which is larger than 0.69 resulting from PRESTO (Supplementary Fig. 11c). These results demonstrated the value of PRESTO in combining the genome-scale coverage of BRENDA with in vivo proteomics chemostat measurements to obtain less biased estimates of $k_{cat}$ values.

**PRESTO with protein-constrained model of *E. coli* metabolism**
To demonstrate the applicability of PRESTO across species, we deployed it with a pcGEM of *E. coli* (eciML1515)[22,26]. To this end, we used a large dataset comprising 31 different growth conditions[12,27–29]. Due to the lack of data on nutrient exchange rates, the same GAM value (i.e., 75.55 $\frac{mmol}{gDWh}$) was used across all conditions. Similarly, we used the same value for total protein content since condition-specific measurements were not available (see the "Methods" section).

By applying three-fold cross-validation, we found the optimal value for the $\lambda$ parameter to be $10^{-5}$ (Supplementary Fig. 12a). This value was associated with an average relative error of 1.95 (average over all $\lambda$: 3.32) and 73 corrected turnover numbers, while on average 156 $k_{cat}$ values were corrected across all explored values for $\lambda$. On average, the Jaccard distance between cross-validation folds was 0.13 (Supplementary Fig. 12b), while the average Jaccard distance between unique sets of enzymes with corrected turnover numbers for each $\lambda$ parameter was three-fold larger (0.4, Supplementary Fig. 12c). Thus, the corrected $k_{cat}$ values among cross-validation folds for each $\lambda$ are more similar (maximum Jaccard distance of 0.29). Moreover, the union of the set of enzymes with corrected $k_{cat}$ values can remain similar over a range of chosen $\lambda$ parameters up to four orders of magnitude (Supplementary Fig. 12c), demonstrating the robustness of the method.

The performance of PRESTO was assessed and compared to GECKO using scenarios (i) and (iii) since no condition-specific uptake rates were available. With default uptake rates, the relative error for predicted growth ranged between 0.01 and 8.56 in the less constrained scenario (i) (Fig. 4a). Further, we obtained relative errors between 0.01 and 0.88 for the more constrained scenario (iii), when using the $k_{cat}$ values corrected by PRESTO (Fig. 4b). In contrast, when using the $k_{cat}$ values from the GECKO approach, the relative error was in the range between 0.01 and the 4.89 for scenario (i) and between 0.89 and 0.99 for scenario (iii). In this scenario, too, we observed that the relative error using $k_{cat}$ values corrected by GECKO was consistently larger

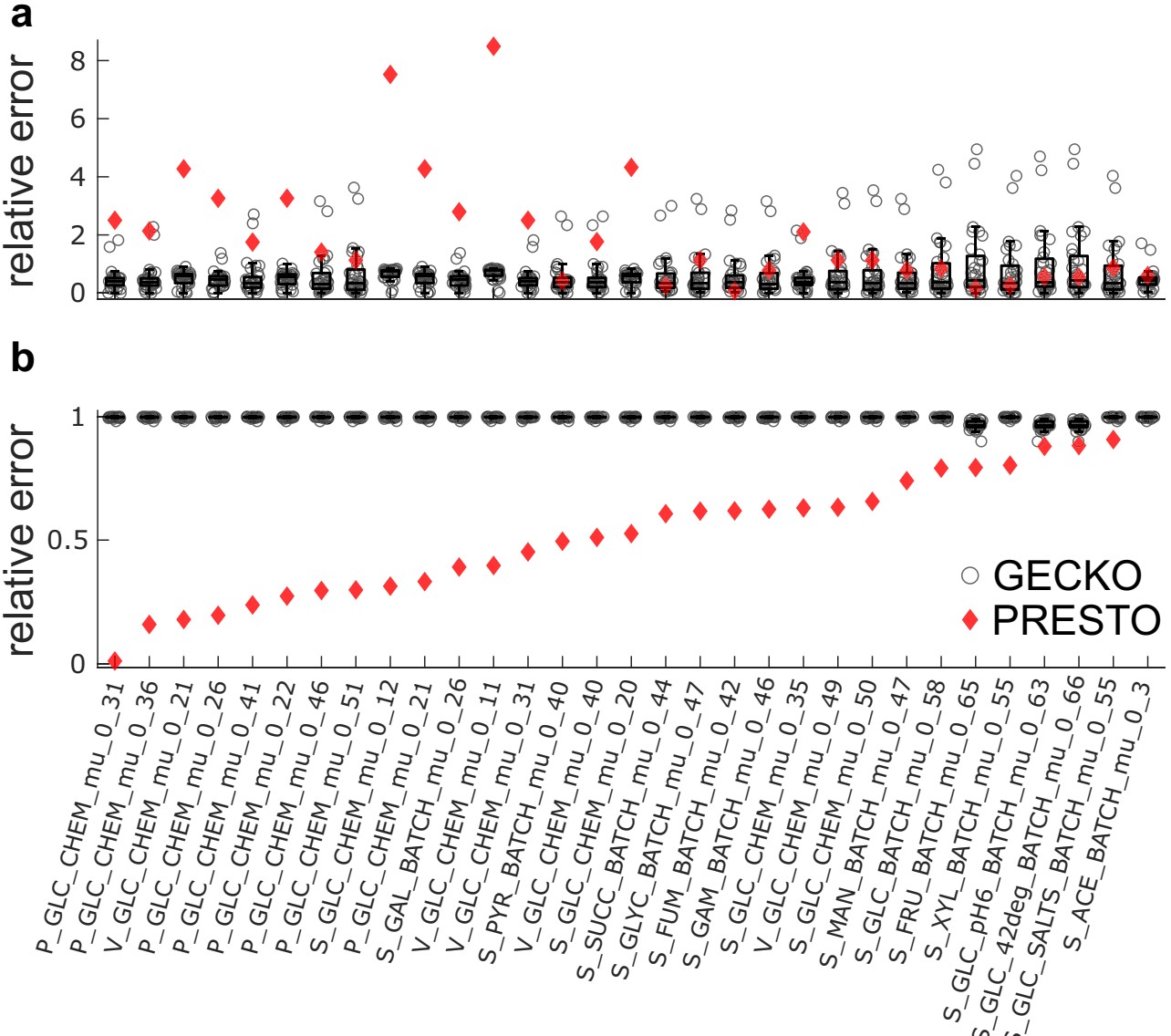

**Fig. 4 | Comparison of predicted growth of *E. coli* from pcGEMs with $k_{cat}$ corrections from GECKO and PRESTO.** Condition-specific pcGEMs with corrected $k_{cat}$ values generated by the GECKO heuristic were used to predict the specific growth rate for each condition (**a**: $n = 31$, **b**: $n = 27$). The boxplots indicate the distribution of the relative error resulting from each set of condition-specific corrected $k_{cat}$ values obtained from the GECKO heuristic. Relative prediction error from each set is indicated by a circle. The red diamonds show the relative errors of predicted specific growth rates from the PRESTO model ($\lambda = 10^{-5}$) by using the single set of corrected $k_{cat}$ values in the respective pcGEM. **a** Only the measured total protein pool was used to constrain the solution and condition-specific uptake rates were bounded by 1000 $\frac{mmol}{h\ gDW}$; **b** abundances of enzymes measured in all conditions were used as additional constraints. Missing data points originate from the infeasibility of the respective models. The compared pcGEMs in each condition used the same respective biomass coefficients, GAM $\sigma$, and $P_{tot}$ values (see the "Methods" section). P: Peebo et al.[28], V: Valgepea et al.[27], S: Schmidt et al.[29]. Middle line and boxes in the box charts in panels **a** and **b** indicate the median and 25th and 75th percentiles, respectively. Outlier values (circles outside the whisker range) are more than 1.5× the interquartile range away from the top or bottom of the box, and whiskers connect the lower or upper quartiles with the non-outlier minimum or maximum. Source data are provided as a Source Data file.

than the relative error resulting from the single set of corrected $k_{cat}$ values obtained by PRESTO (Fig. 4b).

Since we observed high relative errors in the less-constrained scenario (Fig. 4a), we added a second step to PRESTO, which introduces negative corrections that lead to the same relative errors with consideration of proteomics data. This is an optional step that a user can choose to perform in addition to the positive corrections (i.e. relaxation of turnover numbers), introduced in the first step (Supplementary Method 1). As a result, we found 170 negative corrections that reduced the relative error in scenario (i) (Supplementary Fig. 13).

We do not perform a simultaneous search for positive and negative corrections because negative corrections can only reduce the relative error when the current $k_{cat}$ values lead to an overprediction of growth, which is not the case when considering proteomics data. Therefore, no negative corrections are found if the absolute value of introduced positive and negative corrections are to be considered in a single step.

Importantly, the aim of PRESTO is to correct turnover numbers, which represent upper limits on the catalytic efficiency of enzymes. Therefore, we can assume that in vitro turnover numbers that lead to underprediction of specific growth rates when paired with protein abundance data are too low. However, an overprediction of specific growth rates in the same scenario can be caused by thermodynamic, temperature effects, or in-vivo-specific effects. Thus, a reduction of

in vitro turnover numbers results in average apparent catalytic rates for the considered conditions, rather than corrected turnover numbers.

Considering the models with positive $k_{cat}$ corrections, the sum of corrections reached a plateau at $10^{-11}$ for the weighting factor $\lambda$ in the PRESTO objective. We found that the relative cross-validation error at this value was 5.26, which is 2.7-fold larger than the relative error obtained using the optimal $\lambda$. Hence, allowing for more and larger corrections in PRESTO leads to a decrease in the overall relative error within the PRESTO program at the cost of highly biased parameters. The predictions with the highly biased parameters are worse in the test conditions and result in a larger specific growth rate when no enzyme abundance constraints are enforced. This observation is in line with the small number of corrections introduced by the GECKO approach, where only the pool constraint is considered. We conclude that the prediction performance of the eciML1515 model was improved by using turnover numbers corrected by PRESTO only when enzyme abundances are integrated.

To assess the precision of the introduced $k_{cat}$ corrections, we performed variability analysis and sampling (see the "Methods" section) of the introduced corrections to the initial $k_{cat}$ values for two values of the weighting factor $\lambda$, namely $10^{-5}$ and $10^{-11}$. We observed that the 25th and 75th percentiles enclose a narrow interval around the values resulting from PRESTO (Supplementary Fig. 14) and are thus not evenly distributed across the respective interval determined by the variability analysis. We further noted that here, the predictions of smaller $\delta$ are generally more precise than the large corrections ($\delta \geq p_{50}$), which span ~2 orders of magnitude (small $\delta$ ($<p_{50}$): 1.83, Supplementary Fig. 14). However, we also observed that the precision decreased when more corrections were allowed in PRESTO. This further justified our choice for the optimal parameter $\lambda$, which results in a lower number of 73 corrections compared to 204 at $\lambda = 10^{-11}$; moreover, this value guarantees more precise estimates (Supplementary Fig. 17). In conclusion, the application of PRESTO is not limited to a single species but presents a versatile tool for the correction of turnover numbers across species.

In contrast to the observations made in *S. cerevisiae* we found that a model parameterized with in vivo turnover numbers estimated by pFBA[12] outperformed both PRESTO and GECKO in the modeling scenario where no enzyme abundance constraints are taken into account (Supplementary Fig. 11d). This is due to the fact, that pFBA, in contrast to PRESTO and GECKO, can generate estimates lower than the in vitro $k_{cat}$ values, in turn leading to more accurate predictions. However, in the scenario with enzyme abundance constraints, PRESTO predicts specific growth rates closer to the experimental observation in 87% of the conditions (Supplementary Fig. 11e). Thus, in this scenario the integration of information from different modeling conditions achieved in PRESTO serves to obtain $k_{cat}$ value that performs better than the pFBA approach applied by[12].

Finally, we compare the resulting flux distributions and predicted protein abundances by models that are parameterized with $k_{cat}$ values that were corrected using either GECKO or PRESTO. Overall, the feasible ranges ($v^{max} - v^{min}$) for both approaches resulted in Pearson correlation coefficients of 0.985 across all conditions (Supplementary Method 2). The difference between both flux distributions is manifested in a smaller interquartile range in feasible ranges with PRESTO (Supplementary Fig. 15). More specifically, there are fewer reactions with highly constrained flux after introducing $k_{cat}$ corrections with PRESTO compared to GECKO. Moreover, we used the models that were corrected using GECKO and PRESTO to predict protein abundances (Supplementary Method 3), which were then compared to the measured proteomics data using Spearman correlation. Since PRESTO only considers abundances of proteins that were measured across all considered conditions, we computed the correlations for (1) the set of proteins that are measured across all conditions and (2) all protein

abundances per condition. In the first scenario, the correlation was higher after correction with PRESTO in 70% of conditions, compared to the median correlation with GECKO and outperformed all GECKO pcGEMs in 30% of conditions (Supplementary Fig. 16a). When all measured proteins were considered, PRESTO only performed better in 54% of conditions (Supplementary Fig. 16b). Similar to the predicted specific growth rate we also observed better performance for PRESTO models that were subjected to an additional $k_{cat}$ down correction step (Supplementary Method 1). However, we note that the reduced $k_{cat}$ cannot strictly be considered condition-independent $k_{cat}$ values because there may exist physiological states where these enzymes may achieve the efficiency given by the original $k_{cat}$ value (see the "Discussion" section). Since PRESTO considers protein abundances for the correction, which is not the case for GECKO, we expected to find the increased prediction performance with PRESTO compared to GECKO; however, we still observe low overall predictability of protein abundances using the resulting models. Recently, a more sophisticated protein abundance prediction approach using pcGEMs was introduced that can be used to predict more reliable values and might further be improved by considering corrections introduced by PRESTO[30].

Interestingly, in contrast to *S. cerevisiae*, we did not observe larger corrections by PRESTO for organism-unspecific in vitro $k_{cat}$ values (Supplementary Fig. 8b). For the condition-specific sets of $k_{cat}$ corrections introduced by GECKO we observe that all smaller sets were proper subsets of larger sets (Supplementary Fig. 4b) The low number of corrections introduced by GECKO leads to an overlap of only three (75%) enzymes whose $k_{cat}$ values were also corrected by PRESTO (Supplementary Data 3, Supplementary Fig. 18a). These three enzymes catalyze reactions in three distinct metabolic pathways: Phosphoribosylformylglycinamidine synthase acts in the synthesis of purines, while serine acetyltransferase and NADP dependent Ketol-acid reductoisomerase are involved in the synthesis of sulfur amino acids and hydrophic amino acids, respectively (Supplementary Data 4). The pathway enrichment analysis for all PRESTO corrections at $\lambda = 10^{-5}$, indeed also identified amino acid and secondary metabolite synthesis as significantly enriched terms among the enzymes with corrected turnover numbers (Supplementary Fig. 18b). These results argue for a systematic underestimation of in vivo turnover numbers in the pathways in in vitro experiments, irrespective of the investigated organism. However, the lower-order KEGG pathway terms enriched in *E. coli* do not overlap with the ones found in *S. cerevisiae*. Here, fatty acid metabolism and the synthesis of hydrophobic amino acids are among the pathways requiring the correction of turnover numbers.

### Robustness of turnover number corrections

All of the approaches for estimation of in vivo turnover numbers rely on predicted (or estimated) fluxes and protein abundances from multiple conditions[12,14,15], but have not investigated the robustness of the estimates to the number of conditions used. Therefore, next, we investigate the difference in the sets of enzymes with corrected turnover numbers and the concordance of their corrections when ten randomly sampled subcollections of $M$ experimental conditions ($M = 3, 5, 10, 15$) were used instead of all experiments. The differences and concordance were quantified with respect to the estimates obtained by considering data from all available experiments using the Jaccard index and the Pearson correlation coefficient, respectively. In the case of *S. cerevisiae*, we found that the smallest Jaccard difference over 200 scenarios was 0.36, while for *E. coli* this was 0.41 (Supplementary Figs. 19 and 20). In addition, the Pearson correlation coefficient between the (log-transformed) corrected turnover numbers with consideration of all versus a subcollection of $M$ experiments in *S. cerevisiae* ranged from 0.99 to 1.00 (for $M = 15$) to 0.11 and 1.00 (for $M = 3$) (Supplementary Fig. 19). Repeating the analysis in the case of *E. coli*, we found that that the Pearson correlation coefficient ranged from 0.15 to 1.00 (for $M = 15$) to 0.14 and 1.00 (for $M = 3$) (Supplementary Fig. 20).

This is in line with the expectation that the corrections stabilize with an increasing number of experiments. Altogether, these findings pointed out the robustness of turnover number corrections derived from PRESTO with the number of available experiments.

## Discussion

Characterization of enzyme parameters that can inform models of reaction rates is key to expanding and further propelling the usage of metabolic models in diverse biotechnological applications. While the generation of pcGEMs has facilitated the integration of more biophysically relevant constraints, it necessitates access to estimates of turnover numbers as key enzyme parameters. We assessed the bias in the available in vitro and in vivo estimates of turnover numbers as the discrepancy between measured and predicted growth in the ultimate validation scenario when they are combined with constraints from protein abundances. We use the modeling scenario that considers measured protein abundances as the ultimate validation scenario not only for the prediction of metabolic fluxes but also for the prediction of specific growth rates as it contains considerably more biochemically relevant constraints. Indeed for this scenario, we showed that condition-specific growth rates cannot be reliably predicted with pcGEMs of *S. cerevisiae* and *E. coli* when available in vitro (Figs. 2c and 4b) and in vivo estimates (Supplementary Fig. 11c, e) of turnover numbers are used. GECKO resolves this issue by flexibilities measured protein abundances without considering physiological information during the procedure. To overcome this limitation, we developed PRESTO, which corrects turnover numbers and facilitates the integration of enzyme abundance constraints.

In contrast to PRESTO, GECKO uses measured total protein content from a single condition to achieve specific growth rates in the process of correcting the turnover numbers. As a result, the corrected turnover numbers vary between different experiments. Like all existing approaches for the estimation of in vivo turnover numbers based on GEMs, we integrated protein abundance data directly to correct turnover numbers. Following this strategy in PRESTO is further justified by the observation that the turnover numbers included in pcGEMs are often neither from the same enzyme (i.e., EC number), substrate, nor organism. While in vivo turnover number estimates can be adjusted by considering recently proposed Bayesian statistical learning[18], this approach has not considered protein abundance information from proteomics measurements.

We employed PRESTO with the largest data set of these measurements available to date for *S. cerevisiae*[15] and *E. coli*[12]. Through a series of comparative analyses, we demonstrated that the corrections of turnover numbers from PRESTO ultimately increase the prediction accuracy of condition-specific growth for the two organisms when enzyme abundance data are integrated into the corresponding pcGEMs.

Since PRESTO generates a condition-independent $k_{cat}$ set it is bound to correct parameters that lead to the underprediction of biological fluxes. The same reasoning is applied when obtaining in vivo $k_{cat}$ estimates from pFBA by taking the maximum of apparent catalytic rates, and overall conditions[12,15]. Nevertheless, we included an optional step, that additionally allows for the reduction of in vitro $k_{cat}$ values, which can be considered average apparent catalytic rate estimates and result in better performance in the *E. coli* experiments (Supplementary Figs. 13 and 16). Moreover, we showed that in vivo turnover number proxies, obtained by the ranking of condition-specific estimates that use proteomics and fluxomics data, are more highly (but modestly) correlated to estimates from PRESTO than to in vitro turnover numbers. Owing to the constraint-based formulation of PRESTO, we also determined the precision of the correction of turnover numbers. Previous studies have shown that even for the well-studied model organism *Saccharomyces cerevisiae*, only 52% of enzyme turnover numbers in the pcGEM can be

obtained from organism-specific in vitro measurements[22]. Using organism unspecific $k_{cat}$ values for parameterization and correction of pcGEMs, as done in the GECKO pipeline, assumes that enzyme kinetic properties are comparable within one EC number class[31,32]. However, we did not identify clear differences between EC classes, down to the second digit, when considering the distribution of $k_{cat}$ similarities within EC classes (Supplementary Fig. 21). Indeed, it has been reported that EC class plays only a minor role in the prediction of turnover numbers[19] and show stronger similarity with concordant GO categories[33]. Interestingly, our findings show that the turnover number corrections obtained from PRESTO are more precise than EC class-based corrections. (Supplementary Fig. 6). Together, these findings demonstrated PRESTO can be readily used to decrease the bias of turnover numbers. This paves the way for employing the outcome of PRESTO and future extensions toward effectively predicting the kcatome from available protein sequences.

## Methods

### Experimental data

For *S. cerevisiae*, we made use of a dataset gathered by[15] from four different studies[34–37], which included protein abundance data (mmol gDW$^{-1}$) as well as measured growth or dilution rates (h$^{-1}$) and nutrient exchange fluxes (mmol gDW$^{-1}$h$^{-1}$). Exchange fluxes missing in certain conditions were set to 1000 mmol gDW$^{-1}$h$^{-1}$ if the nutrient was present in the used culture media. We further extended this data set by total protein content measurements (g gDW$^{-1}$) from the original studies. For subsequent analyses, we used the maximum abundance of each protein over all replicates per experimental condition. Similarly, we used the average value for specific growth rates and nutrient exchange rates. Since no measurement of total protein content was available for the two conditions evaluated in the Di Bartolomeo study[37], we used the maximum protein content measured across the remaining conditions for these conditions (i.e., 0.67 g/gDW). Moreover, we excluded three temperature stress conditions (i.e., Lahtvee2017_Temp33, Lahtvee2017_Temp36, Lahtvee2017_Temp38) from the analysis since the temperature can have a large effect on the catalytic activity of an enzyme. Gene names in the proteomics dataset were translated to UniProt identifiers using the batch retrieval service of the UniProt REST API[38].

For *E. coli*, we used a dataset comprising 31 experimental conditions, which was gathered by Davidi and colleagues and augmented by Xu et al.[12,14] from three publications[27–29]. Here, too, we used the maximum protein abundance over all replicates (in mmol gD$W^{-1}$). Due to the absence of total protein content measurements in two of the original studies, we relied on the maximum protein content measured in the Valgepea study (i.e., 0.61 g/gDW) to be used for all conditions. Since precise data on nutrient uptake rates were only given for a few conditions, we assigned a default upper bound of 1000 mmol gDW$^{-1}$ h$^{-1}$ to all nutrients contained in the minimal medium (Supplementary Table 2) with additional carbon sources as specified. Gene identifiers were translated to UniProt similar as for *S. cerevisiae*.

### Model preparation

The proposed approach aims at parsimonious correction of turnover values in genome-scale enzyme-constraint metabolic models using measured protein abundances. Therefore, it is important to consider the differential association between enzymes and reactions, i.e., isozymes, enzyme complexes, and promiscuous enzymes. We decided to use the GECKO formalism[7], which deals with these problems elegantly by directly encoding the required information in the stoichiometric matrix. The genome-scale metabolic models for *S. cerevisiae* (YeastGEM v.8.5.0) and *E. coli* (iML1515) were obtained from the yeast-GEM and ecModels GitHub repository,

respectively[22,39] [https://github.com/SysBioChalmers; accessed on 22.08.2021]. For subsequent steps, functions of the COBRA v3.0 toolbox[40] and GECKO2.0 toolbox[22] were employed, of which several functions were adapted for our purposes.

To arrive at raw protein-constrained models for both organisms, the GECKO2.0 model enhancement pipeline was adapted to allow the $k_{cat}$ correction procedure to be omitted. Moreover, any manual corrections of turnover numbers were excluded from model generation. In the process of adapting the raw pcGEM to the respective experimental conditions for both organisms, the GAM value per condition was fitted using the *scaleBioMass* function of GECKO2.0, based solely on the condition-specific nutrient exchange rates, and returning the minimum ($9 \frac{mmol}{gDWh}$) or maximum ($161 \frac{mmol}{gDWh}$) interval boundary if reached (only *S. cerevisiae*). Furthermore, we omitted enzyme abundances, which were not measured across all experiments as the approach proposed here is only applicable for enzymes in the set with measured abundances ($M$).

## PRESTO approach
In the design of PRESTO, we modified the enzyme mass-balance constraints of the augmented stoichiometric matrix, created by GECKO, from

$$-\frac{1}{k_{cat}^{ij}} v_j + e_i = 0 \qquad (1)$$

to inequality constraints that use the measured protein abundance directly. The variable $e$ denotes the predicted protein abundance in the pcGEM, while $E$ represents the vector of measured enzyme abundances. Further, we assume a single turnover number per enzyme $i$ over all catalyzed reactions ($k_{cat_i}^{min} = \arg\min_j k_{cat}^{ij}$):

$$\forall r \in R \sum_{i \in GPR(r)} v_r \le k_{cat_i}^{min} \cdot e_i. \qquad (2)$$

GPR stands for gene–protein-reaction rule that associates reactions ($R$) with underlying genes and proteins. The variable $E$ denotes the measured protein abundance in mmol gDW$^{-1}$. We justify making the assumption for Eq. (2) based on our observation that most enzymes in the *S. cerevisiae* model are associated with no more than four reactions (Supplementary Fig. 22a, c). Further, the vast majority of enzymes are assigned a single unique turnover number even though they catalyze multiple reactions (Supplementary Fig. 22b, d).

We then introduced a correction factor $\delta$, which is added to each $k_{cat}$ if the protein abundances for the underlying enzyme were available:

$$\forall r \in R \sum_{i \in GPR(r)} v_r \le \left( k_{cat_i}^{min} + \delta_i \right) [E_i]. \qquad (3)$$

To find a biologically relevant minimal set of adaptations with respect to the sum of $\delta$, we minimized the weighted sum of the average absolute relative errors, $\omega$, between measured ($\mu^{exp}$) and predicted specific growth rates ($v_{bio}$) overall experimental conditions $C$, and the average $\delta$:

$$\min_{v,\delta,\omega} \frac{1}{|C|} \sum_{j \in C} \omega_j + \frac{\lambda}{|M|} \sum_{i \in M} \delta_i. \qquad (4)$$

Finally, the linear programming formulation of the $k_{cat}$ correction in PRESTO is the following:

$$\min_{v,\delta,\omega} \frac{1}{|C|} \sum_{j \in C} \omega_j + \frac{\lambda}{|M|} \sum_{i \in M} \delta_i$$

subject to

$$Nv^j = 0, \forall j \in C \qquad (5)$$

$$\sum_{i \in GPR(r)} v_r^j \le \left( k_{cat,i}^{min} + \delta_i \right) \left[ E_i^j \right], \forall r \in R, i \in M, \forall j \in C$$

$$v_{min}^j \le v^j \le v_{max}^j; \forall j \in C \qquad (6)$$

$$\delta_i \le (\varepsilon - 1) \cdot k_{cat,i}^{min}, \forall i \in M \qquad (7)$$

$$k_{cat,i}^{min} + \delta_i \le K^{max}, \forall i \in M \qquad (8)$$

$$v_{bio}^j \cdot \omega_j \ge \mu_{exp}^j - v_{bio}^j, \forall j \in C \qquad (9)$$

$$v_{bio}^j \cdot \omega_j \ge v_{bio}^j - \mu_{exp}^j, \forall j \in C \qquad (10)$$

$$\omega \le \theta, \quad \delta \ge 0.$$

The value for $\omega$ was bound from above by a value $\theta$, which was set to 0.6. Constraints that enforce metabolic steady state are captured in Eqs. (5) and (6) represent the lower and upper bounds in the flux through each reaction in each condition, respectively. The constraints in Eqs. (7) and (8) impose an upper bound on $\delta$, which is the minimum of the allowed fold change in $k_{cat}$ values, $\varepsilon$, and a cut-off value $K^{max}$, which denotes the maximum allowed $k_{cat}$ value. The value for $\varepsilon$ was set to $10^5$ since lower values did not yield solutions and $K^{max}$ was set to 57,500,000 s$^{-1}$ (5.3.1.1, *Pyrococcus furiosus*[41]). Equations (9) and (10) ensure that $\omega$ is equal to $\frac{|\mu_j^{exp} - v_{bio,j}|}{v_{bio,j}}$.

The parameter $\lambda$ controls the trade-off between both minimization objectives (see Eq. (4)). As $\lambda$ is unknown and may also be condition- and model-specific, it was fitted using a 3-fold cross-validation scheme, which was repeated for 10 iterations. To this end, we scanned a log-scale interval between $10^{-14}$ and $10^{-1}$. In each iteration, we performed $k_{cat}$ corrections on two folds of condition-specific models and validated the obtained corrections on the remaining fold of condition-specific models. The validation was done by predicting growth only with a constraint on total protein content, without constraints from measured protein abundances. This was done to counteract over-prediction in the scenario without constraints from proteomics data. The relative errors ($\omega$) and the sum of $\delta$ (i.e., $\Delta$) were then used to calculate the scores $s_\lambda$, which helped us choose the optimal value for $\lambda$:

$$s_\lambda = \frac{1}{10} \sum_{\tau=1}^{10} \frac{\omega_{\lambda,\tau} - \omega_{\lambda,\tau}^{min}}{\omega_{\lambda,\tau}^{max} - \omega_{\lambda,\tau}^{min}} \cdot \frac{\log_{10} \frac{\Delta_{\lambda,\tau}}{\Delta_{\lambda,\tau}^{min}}}{\log_{10} \frac{\Delta_{\lambda,\tau}^{max}}{\Delta_{\lambda,\tau}^{min}}}. \qquad (11)$$

The score can be described as the average product of min-max-scaled $\omega$ and $\Delta$ across the 10 cross-validation iterations per explored $\lambda$. The optimal value was then determined by finding the first sign change in the second numerical gradient over $s_\lambda$, starting from the maximum value for $\lambda$. In addition to the optimal $\lambda$, we also compared our results to a second $\lambda$, where the sum of $\delta$ reached a plateau ($\lambda = 10^{-10}$ for *S. cerevisiae* and $\lambda = 10^{-11}$ for *E. coli*, Supplementary Fig. 2a and 12a). The presented approach and analysis scripts were implemented using MATLAB[42].

## Variability analysis for $\delta$

While PRESTO considers multiple experimental conditions to find a set of universal corrections for $k_{cat}$ values, it does not provide an exhaustive view over all possible solutions to this problem. To assess the precision of the corrections, we first performed a variability analysis for $\delta$ to find the minimum and maximum possible values. To guarantee that a solution of equal quality is found with respect to the previously determined sum of $\delta$ and the relative errors to experimentally measured specific growth rates (i.e., $\omega^{opt}$), corresponding constraints were added to arrive at the following linear programming problem:

$$\min_{v,\delta,\omega} / \max_{v,\delta,\omega} \delta_i$$

s.t.

$$Nv^j = 0, \forall j \in C$$

$$\sum_{i \in \text{GPR}(r)} v_r^j \le \left(k_{cat,i}^{min} + \delta_i\right)\left[E_i^j\right], \forall r \in R, i \in M, \forall j \in C$$

$$v_{min}^j \le v^j \le v_{max}^j, \forall j \in C$$

$$\delta_i \le (\varepsilon - 1) \cdot k_{cat,i}^{min}, \forall i \in M$$

$$k_{cat,i}^{min} + \delta_i \le K^{max}, \forall i \in M$$

$$v_{bio}^j \cdot \omega_j \ge \mu_{exp}^j - v_{bio}^j, \forall j \in C$$

$$v_{bio}^j \cdot \omega_j \ge v_{bio}^j - \mu_{exp}^j, \forall j \in C$$

$$0.99 \cdot \omega_j^{opt} \le \omega_j \le 1.01 \cdot \omega^{opt}, \forall j \in C \qquad (12)$$

$$\Delta^{opt} - 10^{-3} \le \sum_{i \in M} \delta_i \le \Delta^{opt} + 10^{-3} \qquad (13)$$

$$\omega \le \theta, \quad \delta \ge 0.$$

The minimal relative error determined for each condition $j$ was fixed within a narrow tolerance ($\pm 1\%$, Eq. (12)) and the minimum sum of corrections $\Delta$ was fixed with a tolerance of $\pm 10^{-3} h^{-1}$ (Eq. (13)).

As the distribution within the obtained min/max intervals can be skewed, we sampled 10,000 points within the obtained intervals. For uniform random sampling, we created random vectors of corrections $\delta^*$ within the determined intervals and projected them onto the solution space by minimizing the distance of $\delta$ to the respective random vector. Therefore, we updated the objective of the program above:

$$\min_{v,\delta,\omega} \sum_{i \in M} |\delta_i - \delta_i^*|. \qquad (14)$$

To ensure reproducibility and compatibility with the COBRA toolbox[40], we solved all optimization problems using the *optimizeCbModel* of the COBRA toolbox. Within this environment, we used the Gurobi solver v9.1.1[43] but we note that any other supported solver can also be used. As we observed numerical instability of the problems in some cases, we decreased the feasibility tolerance (i.e., *feasTol* parameter) for the COBRA solver to $10^{-9}$ for all predictions. The results were visualized using MATLAB[42].

## Validation of corrected models

We used the adapted GECKO pipeline (fitting a condition-specific GAM; excluding manual $k_{cat}$ adaptions) to obtain models with $k_{cat}$ values adapted according to the objective control coefficient heuristic. We note that, when no manual modifications were introduced to the *S. cerevisiae* models, the $k_{cat}$ adaption of the GECKO pipeline would stop because no objective control coefficient above the threshold of 0.001 could be found, and corrected models would still be below the predicted growth error tolerance of 10%. To compare the predictive performance of PRESTO and GECKO corrected models, the models were adapted with the same condition-specific GAM, biomass reaction, and total protein content, $P_{tot.}$ Additionally, PRESTO models were constrained using the same condition-specific saturation rate, $\sigma$ and enzyme mass fraction, $f$, as obtained from the GECKO pipeline. In contrast to the GECKO formulation, we did not subtract the mass of measured enzymes from the total protein pool constraint but instead introduced the measured protein concentration as the upper bound on the enzyme usage reaction, $E_i$, in the respective scenario. This formulation still guarantees that the mass of all used enzymes is lower or equal to the approximated cellular protein pool according to

$$\sum_{i \in r\text{GPR}(r)} e_{i,j} \cdot \text{MW}_i \le P_{tot,j} \cdot f \cdot \sigma_j, \forall j \in C \qquad (15)$$

where MW is the respective molecular weight of the protein. By considering measured and unmeasured enzymes in Eq. (15) we do not have to change $f$ and use the same factor as for the scenario where no protein abundance measures are used[7]. Maximum growth was predicted in three different constraint scenarios: (i) using only the protein pool constraint and default uptake rates (1000 mmol/gDW/h), (ii) using the pool constraint and experimentally measured uptake rates, (iii) using the previous constraints plus the absolute enzyme abundance.

The two studies which generated in vivo $k_{cat}$ values from pFBA[12,15] calculated a single value per reaction irrespective of the presence of isoenzymes. Thus, to parameterize the raw pcGEM (containing only uncorrected BRENDA values) we substituted the in vitro $k_{cat}$ values of all isoenzyme reactions with the respective estimate provided in the study. Reactions catalyzed by complexes were not corrected. Since PRESTO and the pFBA studies provide a single condition-independent model, we generated a condition-independent GECKO model by following the maximum overall conditions approach: For the comparisons, shown in Supplementary Figs. 11 and 15, the condition-wise GECKO models were aggregated into a single union model in which for each reaction the maximum $k_{cat}$ value was used.

## Pathway enrichment analysis

The KEGG pathway terms[44], associated with each enzyme that was measured in all conditions, were acquired using the KEGG REST API. The one-sided p-value, p, for significant enrichment of a pathway term among the enzymes with corrected $k_{cat}$ values was calculated using the hypergeometric density distribution:

$$p(x) = 1 - \sum_{i=1}^{x-1} \frac{\binom{K}{i}\binom{M-K}{N-i}}{\binom{M}{N}}. \qquad (16)$$

Only KEGG pathway terms associated with at least two corrected enzymes were taken into consideration. The p-values associated with all tested pathway terms were corrected for a false discovery rate of 0.05 using the Benjamini–Hochberg correction[45].

## Reporting summary

Further information on research design is available in the Nature Portfolio Reporting Summary linked to this article.

## Data availability

The protein abundance data used in this study have been previously published[12,14,15]. The UniProt database[46] (www.uniprot.org) was used for mapping gene IDs to protein IDs, and the KEGG[44] (www.kegg.jp) database was used to retrieve pathway information for genes. Source data are provided with this paper.

## Code availability

All code that was used to generate the results of this study, including the PRESTO method, are available at GitHub [https://github.com/pwendering/PRESTO] and at Zenodo [https://zenodo.org/record/7675009][47].

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

## Acknowledgements

P.W. and Z.N. would like to thank the Research Focus Group "Evolutionary Systems Biology" of University of Potsdam for funding. Z.N., M.A., and Z.R. would like to thank the Max Planck Society for funding. Z.R. was supported by the European Union's Horizon 2020 research and innovation program grant 862201 (to Z.N.). M.A. and Z.N. were supported by the European Union's Horizon 2020 research and innovation program, project PlantaSYST (SGA-CSA No. 739582 under FPA No. 664620, to Z.N.) (this publication reflects only the author's view and the Commission is not responsible for any use that may be made of the information it contains).

## Author contributions

P.W., M.A. performed research and analyzed data, P.W. contributed code for PRESTO approach, M.A. assessed model performance and performed the statistical analysis, P.W., M.A., Z.R., Z.N. designed research, P.W., M.A., Z.N. wrote the paper.

## Funding

## Competing interests

The authors declare no competing interests.
