## [Peer Review File · Nature Communications]

Data integration across conditions improves turnover number estimates and metabolic predictionsReviewers' Comments:

Reviewer #1:

Remarks to the Author:

Genome-scale metabolic models (GEMs) have been widely used for biological applications. In the past years, protein-constrained GEMs (pcGEMs) have shown better performance compared with traditional GEMs, which however depend on genome-scale turnover numbers. In this manuscript, Wending et al proposed an approach named PRESTO to correct turnover numbers within one of the pcGEM modeling frameworks i.e. GECKO. While the topic is of great interest, there are some issues that need to be addressed.

1 Validity:

1.1 Given that PRESTO attempts to generate a single set of turnover numbers expected to fit all conditions, the authors seem to assume that turnover numbers are unchanged across conditions, which is obviously incorrect as the turnover numbers are very condition dependent. Therefore the assumption behind PRESTO is not sound.

1.2 For the simulations without enzyme abundance constraints, PRESTO even performed worse than GECKO in most cases (Fig2 and Fig4), which could be explained by the invalid assumption mentioned above. These results suggest that there is no definite improvement using PRESTO.

1.3 The authors stated that the turnover numbers need to be corrected due to underestimation (line 178). Does it mean that the correction of all the turnover numbers using PRESTO is increase, no decrease? It is a serious flaw of PRESTO if it can only increase the turnover numbers. What if the initial turnover numbers are large enough?

1.4 The section "Robustness of turnover number corrections" looks very interesting. The authors showed that the corrections became robust with the number of available experiments. In the *S. cerevisiae* case the corrections stabilized after $M = 10$ or 15 , but in the *E. coli* case they did not yet stabilize even at $M = 15$. Do the results suggest that PRESTO requires as many (at least 10) experiments as possible? If so, PRESTO would then be limited to organisms with rich data.

1.5 It is not surprising that more experiments used for training result in more robust corrections, but there comes another important question: do more experiments really result in better predictions? This is again related to comment 1.1.

While in the *S. cerevisiae* case $M = 3$ showed a wide variation (FigS15b), which seems to be not robust, it is very likely that those clustered circles under $\gamma = 0.2$ could predict better than the others for specific conditions if the authors could show the comparison of prediction performance.

2 Limitations:

2.1 PRESTO has limited applications as it can only be used in organisms with rich data (see comment 1.4), i.e. measurements from multiple conditions. Furthermore, PRESTO requires absolute protein abundance data, which are much less available than physiological data.

2.2 PRESTO has a limited improvement (see comment 1.2). It performed better than GECKO only in the cases with constraints of protein abundances, which are again not so commonly measured as physiological data.

2.3 PRESTO appears to be limited to the GECKO formalism. While PRESTO was developed on GECKO, there are different approaches besides GECKO as the authors introduced (lines 49 and 50). Can the authors discuss the potential applications of PRESTO to the other modeling frameworks?

3 Inappropriate conclusions:

3.1 The conclusion that the obtained estimates are more precise than the available in vivo turnover

numbers is not valid (line 22). While the authors demonstrated (lines 204 to 206) that in vivo turnover numbers performed worse than those corrected by PRESTO and GECKO in predicting growth, the previous study (ref 15) showed that in vivo turnover numbers performed better in predicting protein abundances. The better growth predictions by PRESTO and GECKO could be explained by the fact that these two approaches fit with growth data. It is very necessary to test whether the turnover numbers corrected by PRESTO and GECKO can perform better than in vivo turnover numbers in predicting proteomics data. Furthermore, in the E. coli case the authors even found the negative result that the turnover numbers estimated by pFBA outperformed both PRESTO and GECKO as concluded in line 260.

3.2 Lines 141 and 142: these results can only demonstrate that PRESTO outperformed GECKO in simulations with proteomics integration.

3.3 The conclusion in line 235 is not correct according to Fig4a.

4 Others:

4.1 It is strange that the relative errors in Fig4a can be greater than 1 while the others in Fig2abc and Fig4b are all in the range between 0 and 1.

4.2 In FigS2 there is only one dashed vertical line but the authors mentioned a right one and a left one.

4.3 Reference misuse

Line 43: ref 13 is not for *S. cerevisiae*.

Line 50: ref 2 should go to ME-models.

4.4 Typos

Line 169: "then" to "than"

Reviewer #2:

Remarks to the Author:

The paper provides an interesting and novel approach to fine tune K_{cat} values for constraint-based models with protein constraints. It uses an optimization procedure to fix K_{cat} values, to minimize errors in predicted growth rates, also seeking to minimize the changes through a regularization parameter.

The methods is tested with data from yeast and E. coli, with some relevant results.

The paper is well structured and written. The proposed method is generally well explained and technically sound. The results are consistent and well described.

My only concern/ doubt relies on the validation process. On the one hand, it seems that the methods relies solely on growth rates, disregarding other predicted fluxes, which might be limitative. On the other hand, I had some doubts on the cross validation process. There seems to be one level for the estimation of the overall results, and another inner level to estimate the regularization parameter, but it is not clear how this is done and if it is done in an unbiased way. Finally, I have some doubts that given the limitation on the number of conditions used, the results can be so convincing. I believe these issues should be clarified, and the limitations discussed in the conclusions.

Regarding the explanation of the method, expression (11) should be better explained.

Some minor suggested corrections and typos:

- l.35 - add "the" after "due to"

- l. 115 and 493, 494 - "constrain(s)" -> "constraint(s)"

- l. 346 - "of the" repeated

- l. 416 and 418 - In the expressions should be j belongs to C (and not conditions) ?

- l. 482 - "where" -> were

Reviewer #3:

Remarks to the Author:

Genome-scale metabolic models (GEMs) based on constraints have resulted in more accurate prediction of maximum specific growth rates on various carbon sources, flux distributions, and other complicated phenotypes. The construction of protein constrained (pc) GEMs, on the other hand, is critically dependent on the integration of organism-specific enzyme turnover numbers, k_{cat} , which comprise an organism's k_{catome} . Philipp et al. have claimed that the available in vitro and in vivo turnover numbers are biased and lead to poor prediction of condition-specific growth rates with protein-constrained models of *Escherichia coli* and *Saccharomyces cerevisiae*.

Therefore, the authors propose PRESTO to correct turnover numbers by using the respective protein abundances, matching the predictions from pcGEMs with measurements of cellular phenotypes. They demonstrate that predictions of growth by pcGEMs of *S. cerevisiae* with turnover numbers corrected by PRESTO are more accurate than the GECKO model. Significant improvement in the performance of growth prediction is observed and the authors have also demonstrated that the proposed solution is robust. However, some issues need to be clarified, as detailed below.

Major

In lines 57/58, the authors claim that the estimates of K_{cat} values have bias. This is a very strong claim. The authors should provide strong proof and justifications for this claim, as this would go against several published papers on the topic.

The authors have estimated the relative error of the predicted specific growth rate for each condition using flux balance analysis with the pcGEM. To ensure the quality of the final model, the authors should evaluate the turnover number corrected pcGEM using MEMOTE or other metabolic model tests.

The authors have proposed a protein abundance-based correction of turnover number however, they have not specified any requirements/units for protein abundance data. Does the proteomic data have to be in a specific unit (eg. mmol/gDW)? Authors need to provide more details on proteomic data, so that other researchers can use the method.

In lines 357/358, the authors mention that they further augmented the data. More details should be provided to assess this step.

In lines 425 and 426 (equations 9 and 10), the authors have constrained the relative error (ω_j). These two equations seem to contradict each other and need to be clearly explained.

The authors should clearly explain and justify all the constraints applied to solve PRESTO.

In line 428, the authors have set the value of ε to $\{10\}^5$ but have not explained how they chose this value. What should be the value of ε when constructing the pcGEM for other organisms?

The authors have not specified if the correction of turnover number will have any impact on the flux variability. The authors should perform a comparison of flux variability range between PRESTO and GECKO to understand the impact of turnover number correction in flux ranges.

In line 43, the authors mention that in vivo turnover numbers for *S. cerevisiae* and *A. thaliana* do not reflect in vitro measurements. Is this statement true for other organisms? To justify the proposed solution for turnover number correction, it would be better to clarify if the turnover number needs correction for other organisms too.

Can PRESTO be used to correct turnover numbers for other organisms too? If yes, then what are the parameters that need to be taken into consideration while using PRESTO for other organisms like homo sapiens, Mus musculus, etc.?

The source code provided in the github repository is not able to replicate the results in the paper without error. For instance, while running `correct_kcats_yeast.m`, there was an error at line 112. (Error using save. Variable 'deltaSamplingMat' not found).

Minor

Please check the typo in line 392 (returning the minimum)

The notation for relative error in lines 436 and 414 are er and ω respectively. Can you please clarify what these notations mean and how they are different?

Response letter to the Reviewers' comment on

“Data integration across conditions improves turnover number estimates and metabolic predictions”

All original comments are presented in regular font, while our point-by-point responses appear in bold font.

Reviewer #1 (Remarks to the Author):

Genome-scale metabolic models (GEMs) have been widely used for biological applications. In the past years, protein-constrained GEMs (pcGEMs) have shown better performance compared with traditional GEMs, which however depend on genome-scale turnover numbers. In this manuscript, Wending et al proposed an approach named PRESTO to correct turnover numbers within one of the pcGEM modeling frameworks i.e. GECKO. While the topic is of great interest, there are some issues that need to be addressed.

1 Validity:

1.1 Given that PRESTO attempts to generate a single set of turnover numbers expected to fit all conditions, the authors seem to assume that turnover numbers are unchanged across conditions, which is obviously incorrect as the turnover numbers are very condition dependent. Therefore the assumption behind PRESTO is not sound.

We thank the reviewer for pointing out this important aspect in the estimation of (apparent) turnover numbers / catalytic rates. Turnover numbers (kcat), which are measured *in vitro*, represent the maximum number of conversions of a substrate per unit time at a given enzyme concentration (at substrate saturation). Thus, kcat values provide an upper limit on the efficiency of an enzyme with a specific substrate. The kcat values obtained from *in vitro* assays are influenced by the pH value and temperature used in the assay. Since there is no selection for standardized conditions in the generation of a GECKO model, the values can show variability due to assay conditions (e.g. , as well as buffers and cofactors used) besides other factors in the kcat matching process.

In vivo, several additional factors influence the catalytic efficiency of an enzyme, leading to condition-specific apparent catalytic rates (kapp), which are lower than the kcat value. The most prominent factors are, in addition to pH value and temperature, post-translational modifications, substrate saturation, or allosteric inhibition.

Since (sub)cellular pH is tightly regulated (Putnam (2012), <https://doi.org/10.1016/B978-0-12-387738-3.00017-2>), we assume the pH-value in the cytosol and organelles to be comparable across all conditions used in our study. Moreover, we argue that temperature does not play a role in our estimation procedure since we excluded experiments that were carried out at varying temperatures. That said, we note that the estimates resulting from PRESTO should be corrected for temperature effects, if they are to be used in such modelling scenarios (Li et al. (2021) Nat Commun, Wending & Nikoloski (in review)).

Further, we are aware that post-translational modifications (PTM) of enzymes can potentially have an influence on kapp. PTMs of different types have been documented across many enzymes, but the functional role of PTMs on catalytic rates and efficiencies remain scarce (see for instance, <https://www.ncbi.nlm.nih.gov/pmc/articles/PMC5705920/>, <https://pubs.acs.org/doi/pdf/10.1021/acs.biochem.1c00145>, [https://www.jbc.org/article/S0021-9258\(20\)00201-X/fulltext](https://www.jbc.org/article/S0021-9258(20)00201-X/fulltext)) we are not aware of any conclusive genome-wide evidence for the effect

of PTMs on enzyme catalytic rates and efficiency. Since PRESTO does the correction of all turnover numbers (associated with reactions with measured enzyme abundance), the corrected values can be regarded as estimates that best mimic cellular physiology (including specific growth and exchange rates).

Substrate saturation has been shown to cause large differences in apparent catalytic rates between experimental conditions in *S. cerevisiae*. However, PRESTO estimates corrections for k_{cat} values, not apparent catalytic rates. Hence, the correction obtained using PRESTO represents the maximum possible number of conversions per time at substrate saturation. This still allows for lower reaction rates in individual conditions, which may be caused by effects of substrate saturation and other factors mentioned above, such as allosteric inhibition. We updated several passages in the manuscript to clarify the aim of correcting k_{cat} parameters in contrast to the calculation of apparent catalytic rates.

The availability of k_{cat} values is important for the parametrization of large-scale kinetic models (Khodayari & Maranas (2016) Nat Commun, <https://doi.org/10.1038/ncomms13806>), because they present a single upper limit for enzyme efficiency in comparison to condition-specific apparent catalytic rates. Apparent catalytic rates can then be obtained by additional fitting steps considering the effects of enzyme activity and substrate saturation.

For completeness, we note that there are currently two types of approaches available that provide estimates for k_{cat} values. The first type is based on parsimonious FBA, whereby k_{app} values are calculate from the predicted flux distribution for a given condition and measured protein abundances (Davidi et al. 2016 (<https://doi.org/10.1073/pnas.1514240113>), Chen et al. 2021 (<https://doi.org/10.1073/pnas.2108391118>), Xu et al. 2021 (<https://doi.org/10.1093/bioinformatics/btab575>), Arend et al. 2022 (<https://doi.org/10.1101/2022.11.06.515318>)). The maximum k_{app} value over all conditions then serves as a proxy for the k_{cat} value. We compare our results to the results of these kinds of estimation approaches and showed increased performance of PRESTO. The second type is based on machine learning, which however, requires an organism-specific training set of k_{cat} values and does not incorporate proteomics data, which is why we excluded these approaches from our comparison (Heckmann et al. 2018 (<https://doi.org/10.1038/s41467-018-07652-6>), Li et al. 2021 (<https://doi.org/10.1038/s41929-022-00798-z>)).

1.2 For the simulations without enzyme abundance constraints, PRESTO even performed worse than GECKO in most cases (Fig2 and Fig4), which could be explained by the invalid assumption mentioned above. These results suggest that there is no definite improvement using PRESTO.

Given the arguments of the reviewer in the first comment, we note that GECKO predicts very limited condition-specificity. The forward-like regression approach employed by GECKO finds highly overlapping sets of corrections regardless of the condition as we show in the new Figure S4. This can be observed particularly in the case of *E. coli* where the sets of corrected reactions become larger over considered conditions (over a specific order shown in Figure S4).

We further argue that the testing scenario with constraints from proteomics data is the most relevant one, since it contains two orders more biological observations (namely of protein abundances) that can be used as physiological constraints. Scenarios (i) and (ii) were included for full disclosure of model performances and fair comparison to GECKO corrections, but in our opinion,

they are insufficient to assess the quality of more than 1000 k_{cat} parameters, since only media composition or single uptake rates are used as biological information input.

The difference in performance of both approaches in the scenarios with and without constraints from protein abundances is expected since GECKO correction strategy is validated without proteomics constraints, while PRESTO uses proteomics constraints. However, strikingly, the performance with k_{cat} corrections introduced using GECKO is overall poor when constraints from proteomics data are applied, with relative errors close to one. In contrast, models with k_{cat} corrected using PRESTO still show good performance in the two other testing scenarios (i and ii), which means that the relative error is lower or within the interquartile range of relative errors across the “condition-specific” corrections introduced by GECKO. More precisely, only 8 out of 27 models result in relative errors above the 75% quartile for *S. cerevisiae*, while there are 20 out of 31 for *E. coli*.

1.3 The authors stated that the turnover numbers need to be corrected due to underestimation (line 178). Does it mean that the correction of all the turnover numbers using PRESTO is increase, no decrease? It is a serious flaw of PRESTO if it can only increase the turnover numbers. What if the initial turnover numbers are large enough?

Since k_{cat} value are by definition the maximum possible number of substrate conversions by an enzyme per time, we only attempt to correct possible k_{cat} misannotations that led to a k_{cat} value that is too low to explain observed specific growth rates, exchange fluxes and protein abundances. Please note that k_{cat} misannotations can occur easily due to the matching procedure applied by GECKO. As we show in Figure S21, there is also a high variability within second level EC classes, which can lead to lower k_{cat} values even if two EC number digits are matched. Additional sources for k_{cat} misannotations are k_{cat} values from taxonomically distant organisms missing substrate matches.

If initial k_{cat} values are large enough to support experimental data, the k_{cat} values remains unchanged. Prompted by the reviewer’s suggestion, we added an optional second step to our PRESTO method that searches for negative corrections for k_{cat} values that can potentially further decrease the relative error. To this end, we fixed the positive corrections obtained in the first step as well as update fluxes for all conditions. Further, we applied an upper bound on the sum of relative errors. As a result, we obtained 170 negative corrections for the *E. coli* eciML1515. The resulting model was validated in the same way as in Figure 4 in the main text (see new Figure S13). As before, we tested the performance with and without inclusion of proteomics data. We observed that the set of reducing k_{cat} valued decreased the relative error to measured specific growth rates in comparison to only including positive corrections in the first scenario (Figure S13a). However, it was not possible to reduce the relative error when proteomics data were considered (Figure S13b). In contrast, the relative error was increased for two conditions. Notably, the relative errors resulting from PRESTO were identical for the second scenario with and without the second (reduction of k_{cat} values) step. The increased relative errors for the two scenarios can be explained by the protein pool, used as an additional constraint in the validation, but not in PRESTO itself (see the Method formulation).

We thank the reviewer for this comment and updated the manuscript to emphasize that these negative corrections do not necessarily represent k_{cat} values but rather average apparent catalytic rates across all conditions. There may still exist conditions where the enzymes with negative corrections indeed show the k_{cat} value that was initially included in the model.

1.4 The section “Robustness of turnover number corrections” looks very interesting. The authors showed that the corrections became robust with the number of available experiments. In the *S. cerevisiae* case the corrections stabilized after $M = 10$ or 15 , but in the *E. coli* case they did not yet stabilize even at $M = 15$. Do the results suggest that PRESTO requires as many (at least 10) experiments as possible? If so, PRESTO would then be limited to organisms with rich data.

The reviewer is correct – in case of *E. coli*, more than 15 conditions are required until the estimated corrections stabilize. Since this is not the case for *S. cerevisiae*, it means that one will have to investigate the robustness of the corrections for another organism to assess, whether they are stable. The availability of rich data is unfortunately a limitation of all kcat estimation approaches, detailed in the last part of our response to the reviewer’s point 1.1., above. Notably, the aspect of robustness in the estimation of maximum kapp values has not yet been investigated. It would therefore be interesting to study the influence of using subsets of conditions for the approximation of kcat values by maximum kapp values obtained by the approaches based on parsimonious FBA, direction in which we have made first steps (Arend et al. (2022), <https://doi.org/10.1101/2022.11.06.515318>).

1.5 It is not surprising that more experiments used for training result in more robust corrections, but there comes another important question: do more experiments really result in better predictions? This is again related to comment 1.1.

While in the *S. cerevisiae* case $M = 3$ showed a wide variation (FigS15b), which seems to be not robust, it is very likely that those clustered circles under $\gamma = 0.2$ could predict better than the others for specific conditions if the authors could show the comparison of prediction performance.

It could very well be that one of the correction sets obtained with only three models simultaneously performs better for a specific subset of conditions. While these corrections will result in good performance in these few conditions, they will perform much worse in other conditions, i.e. they will be highly biased (indicated by Jaccard index and Pearson correlation to the solution obtained with all conditions (Figures S19 and S20). As mentioned in our response to point 1.1, we are interested in estimating corrections for kcat values that apply for all conditions, and not apparent catalytic rates. Even if we were to select a set of three conditions that shows better corrections for a selected subset of conditions, we do not see a way of selecting another such set of size three in an unbiased way. Hence, while the overall solution using all conditions (or e.g. 15 conditions for *S. cerevisiae*) may not result the lowest possible relative error for all conditions, it provides us with a robust estimation of corrections for the considered kcat values.

2 Limitations:

2.1 PRESTO has limited applications as it can only be used in organisms with rich data (see comment 1.4), i.e. measurements from multiple conditions. Furthermore, PRESTO requires absolute protein abundance data, which are much less available than physiological data.

We kindly direct the reviewer to our response to point 1.4, above.

2.2 PRESTO has a limited improvement (see comment 1.2). It performed better than GECKO only in the cases with constraints of protein abundances, which are again not so commonly measured as physiological data.

We kindly direct the reviewer to our response to point 1.2, above. As previously stated, we consider the case with constraints from protein abundances as the most relevant scenario because it includes most experimental observations. The other testing scenario(s) without consideration of protein abundances are commonly used modelling scenarios and serve as the basis for kcat correction in GECKO. Therefore, we included them for a full disclosure of the performance of the model(s) with corrected kcat values.

2.3 PRESTO appears to be limited to the GECKO formalism. While PRESTO was developed on GECKO, there are different approaches besides GECKO as the authors introduced (lines 49 and 50). Can the authors discuss the potential applications of PRESTO to the other modeling frameworks?

PRESTO can certainly be re-formulated to be applied with other modelling frameworks, e.g. MOMENT or sMOMENT. We chose to use the GECKO formalism as the GECKO toolbox provides a user-friendly framework to generate enzyme-constrained models. Further, all other approaches that generate enzyme-constrained metabolic models, although differently formulated, can be cast into one another and they result in the identical feasible space. Therefore, the application of PRESTO with other modeling frameworks are self-evident and we opted not to include this point in the revised manuscript.

3 Inappropriate conclusions:

3.1 The conclusion that the obtained estimates are more precise than the available *in vivo* turnover numbers is not valid (line 22). While the authors demonstrated (lines 204 to 206) that *in vivo* turnover numbers performed worse than those corrected by PRESTO and GECKO in predicting growth, the previous study (ref 15) showed that *in vivo* turnover numbers performed better in predicting protein abundances. The better growth predictions by PRESTO and GECKO could be explained by the fact that these two approaches fit with growth data. It is very necessary to test whether the turnover numbers corrected by PRESTO and GECKO can perform better than *in vivo* turnover numbers in predicting proteomics data. Furthermore, in the *E. coli* case the authors even found the negative result that the turnover numbers estimated by pFBA outperformed both PRESTO and GECKO as concluded in line 260.

We thank the reviewer for pointing out our mistake in writing that “the obtained estimates are more precise than the available *in vivo* turnover numbers” (L22). The sentence refers to our finding shown in Figure S6, where we show that the root squared error for kcat values associated with the same EC number is on average lower in the corrections introduced by PRESTO than in the BRENDA database. Hence, the term “*in vivo*” was exchanged with “*in vitro*” in the updated version of the manuscript.

We concluded that turnover numbers estimated using pFBA approaches outperformed PRESTO and GECKO in the scenario that only includes the upper bound on the total protein content for the *E. coli* eciML1515 model. However, both the GECKO and pFBA approach performed worse in the scenario

with additional constraints derived from measured protein abundances, except for four conditions. For an explanation of why we consider the scenario with constraints from proteomics data the most relevant one, please refer to point 1.2, above.

We further thank the reviewer for making an excellent point about comparing predictions of protein abundances with the approaches that we compare. In the updated version of the manuscript, we include the additional Figure S16 that shows the Spearman correlation between measured protein abundances and those predicted using GECKO models with k_{cat} values corrected using GECKO or PRESTO. The results show that the updated model after applying PRESTO performed better in predicting protein abundances than the median of GECKO-corrected models in 54% of the conditions when only those proteins were considered that were measured across all conditions. As expected, this performance was improved when PRESTO corrections also allowed reduction in k_{cat} values (using the optional, second step in the updated procedure); however, we note that these estimates cannot be considered k_{cat} values (see point 1.3 for detailed elaboration).

3.2 Lines 141 and 142: these results can only demonstrate that PRESTO outperformed GECKO in simulations with proteomics integration.

We thank the reviewer for this valid point and adapted the respective passage.

3.3 The conclusion in line 235 is not correct according to Fig4a.

We thank the reviewer for pointing this out. We removed the reference to Figure 4a in the updated version of the manuscript.

4 Others:

4.1 It is strange that the relative errors in Fig4a can be greater than 1 while the others in Fig2abc and Fig4b are all in the range between 0 and 1.

After the correction of k_{cat} values by GECKO or PRESTO, the specific growth rates predicted using the *S. cerevisiae* pcGEMs are still lower than the experimentally determined values in all three scenarios. With the *E. coli* model, we observed that corrections introduced by PRESTO often lead to overpredictions of specific growth rates. Please note that this is not the case in the scenario with constraints from proteomics data, which we consider as the most relevant validation scenario. We attempt to counteract this effect by validating the model performance during cross-validation using the scenario with only the constraint on total protein content. The other scenarios were included for a full disclosure of the performance of the corrected models as they present commonly used modelling scenarios.

4.2 In FigS2 there is only one dashed vertical line but the authors mentioned a right one and a left one. **There are two black dashed vertical lines in Figure S2a at x-positions 10^{-10} and 10^{-7} .**

4.3 Reference misuse

Line 43: ref 13 is not for *S. cerevisiae*.

We removed reference 13 from L43.

Line 50: ref 2 should go to ME-models.

We updated the references.

4.4 Typos

Line 169: “then” to “than”

We updated L169.

Reviewer #2 (Remarks to the Author):

The paper provides an interesting and novel approach to fine tune K_{cat} values for constraint-based models with protein constraints. It uses an optimization procedure to fix K_{cat} values, to minimize errors in predicted growth rates, also seeking to minimize the changes through a regularization parameter.

The methods is tested with data from yeast and *E. coli*, with some relevant results. The paper is well structured and written. The proposed method is generally well explained and technically sound. The results are consistent and well described.

My only concern/ doubt relies on the validation process. On the one hand, it seems that the methods relies solely on growth rates, disregarding other predicted fluxes, which might be limitative.

We thank the reviewer for the generally positive feedback. Regarding the validation process, we note that specific growth rates represent key predictions from FBA. Therefore, we decided to include the relative error between predicted and experimentally determined specific growth rates in the objective of PRESTO. However, the objective can easily be changed or updated to include other measured fluxes. With regards to including other measured fluxes, we used measured exchange fluxes as constraints for the individual conditions. Hence, they enter the prediction as constraints but do not appear in the objective function (since they are fixed to respective condition-specific measurements).

On the other hand, I had some doubts on the cross validation process. There seems to be one level for the estimation of the overall results, and another inner level to estimate the regularization parameter, but it is not clear how this is done and if it is done in an unbiased way. Finally, I have some doubts that given the limitation on the number of conditions used, the results can be so convincing. I believe these issues should be clarified, and the limitations discussed in the conclusions.

The regularization parameter λ controls the trade-off between the minimization of relative errors to measured specific growth rates and the sum of corrections of the k_{cat} values. As for other approaches that require hyperparameter tuning, e.g. LASSO, we find λ in an unbiased way by performing cross-validation. We performed 3-fold cross-validation, with 50 iterations for values λ between 10^{-14} and 10^{-1} and saved (1) relative errors obtained by applying only a constraint on the total protein content without any constraints from proteomics data, and (2) the sums of introduced corrections. We used these values to arrive at scores for each explored λ (Equation 12). By taking the second numerical gradient of the scores over all λ , we find the optimal λ at the first sign change from the right (the direction of the largest to the smallest λ). We have updated the paragraph around Eq. 12 to include more detailed information on the cross-validation.

We agree with the reviewer, that additional data – especially in the case of *E. coli* – will likely lead to even more accurate k_{cat} values obtained from PRESTO, and novel measurements can easily be incorporated into PRESTO once they become available. We are convinced that using the largest data set of absolute proteomics measurements available to date, already leads to considerable improvements in the model parameterization and the cross-validation approach employed by PRESTO leads to less biased parameter sets than comparable pFBA based approaches.

Regarding the explanation of the method, expression (11) should be better explained.

We added an explanation in the updated version of the manuscript.

Some minor suggested corrections and typos:

- l. 35 - add "the" after "due to"
- l. 115 and 493, 494 - "constrain(s)" -> "constraint(s)"
- l. 346 - "of the" repeated
- l. 416 and 418 - In the expressions should be j belongs to C (and not conditions) ?
- l. 482 - "where" -> were

We addressed all points in the updated version.

Reviewer #3 (Remarks to the Author):

Genome-scale metabolic models (GEMs) based on constraints have resulted in more accurate prediction of maximum specific growth rates on various carbon sources, flux distributions, and other complicated phenotypes. The construction of protein constrained (pc) GEMs, on the other hand, is critically dependent on the integration of organism-specific enzyme turnover numbers, *k_{cat}*, which comprise an organism's *k_{cat}*ome. Philipp et al. have claimed that the available *in vitro* and *in vivo* turnover numbers are biased and lead to poor prediction of condition-specific growth rates with protein-constrained models of *Escherichia coli* and *Saccharomyces cerevisiae*.

Therefore, the authors propose PRESTO to correct turnover numbers by using the respective protein abundances, matching the predictions from pcGEMs with measurements of cellular phenotypes. They demonstrate that predictions of growth by pcGEMs of *S. cerevisiae* with turnover numbers corrected by PRESTO are more accurate than the GECKO model. Significant improvement in the performance of growth prediction is observed and the authors have also demonstrated that the proposed solution is robust. However, some issues need to be clarified, as detailed below.

Major

In lines 57/58, the authors claim that the estimates of *K_{cat}* values have bias. This is a very strong claim. The authors should provide strong proof and justifications for this claim, as this would go against several published papers on the topic.

We thank the reviewer for the critical review of the assumption of the proposed approach. When we claim that estimated *in vivo* turnover numbers are biased, we do so with regards to the observed prediction performance of specific growth rates with constraints from experimentally measured protein abundances and exchange fluxes. We consider the testing scenario with constraints from proteomics data the most relevant validation scenario as it includes the most experimental data.

The authors have estimated the relative error of the predicted specific growth rate for each condition using flux balance analysis with the pcGEM. To ensure the quality of the final model, the authors should evaluate the turnover number corrected pcGEM using MEMOTE or other metabolic model tests.

We thank the reviewer for suggesting the usage of external validation methods. Please note that MEMOTE does not support analysis of pcGEM parameter. The only differences between the original model and the model resulting from PRESTO are the updated *k_{cat}* values, which do not alter any topological properties of the network (which would be relevant for analysis with MEMOTE). Additional quality assessment can only be done by considering additional experimental data, which are currently not available for a comparable number of conditions.

The authors have proposed a protein abundance-based correction of turnover number however, they have not specified any requirements/units for protein abundance data. Does the proteomic data have to be in a specific unit (eg. mmol/gDW)? Authors need to provide more details on proteomic data, so that other researchers can use the method.

We would like to thank to reviewer for this suggestion. We updated the methods section to specify the required unit for protein abundances, which is mmol/gDW.

In lines 357/358, the authors mention that they further augmented the data. More details should be provided to assess this step.

We added a sentence in the updated version of the manuscript.

In lines 425 and 426 (equations 9 and 10), the authors have constrained the relative error (ω_j). These two equations seem to contradict each other and need to be clearly explained.

The constraints in Eq. 10 and 11 are used to define the relative error, ω , as the absolute difference between the measured and predicted specific growth rate. If the difference between μ_j^{exp} and $v_{bio,j}$ is non-zero, either the right side of Eq. 10 or the right side of Eq. 11 will be negative. Because ω is the absolute relative error, it must be at least as large as $\frac{|\mu_j^{exp} - v_{bio,j}|}{v_{bio,j}}$, which is what these constraints achieve.

We added a more detailed explanation of the constraints in the updated version of the manuscript.

The authors should clearly explain and justify all the constraints applied to solve PRESTO.

We added a more detailed explanation of the constraints in the updated version of the manuscript.

In line 428, the authors have set the value of ε to $\{10\}^5$ but have not explained how they chose this value. What should be the value of ε when constructing the pcGEM for other organisms?

We thank the reviewer for this excellent point. We found the parameter $\varepsilon = 10^{-5}$ by applying our approach with different, smaller, values for ε , and $\varepsilon = 10^{-5}$ is the smallest fold-change that allowed a feasible solution (with the maximum allowed relative error at 0.6) for both models. We used the same value for both models to keep the space of possible parameter value combinations as low as possible. We suggest using a value of $\varepsilon = 10^{-5}$ for other organisms as well, and it should only be increased in case it leads to infeasibility. Nevertheless, when there is no upper limit imposed on the relative error, there is also no lower limit for ε . In this case, one would also have to tune ε as a hyperparameter using grid-like cross-validation.

The authors have not specified if the correction of turnover number will have any impact on the flux variability. The authors should perform a comparison of flux variability range between PRESTO and GECKO to understand the impact of turnover number correction in flux ranges.

We thank the reviewer for the suggestion. We added Figure S15 that shows a comparison of feasible ranges after kcat correction for the eciML1515 model using GECKO and PRESTO, respectively. This

analysis showed that the corrections introduced by PRESTO do not affect the median flux range but increase the 25% percentile of the distribution of feasible ranges and slightly decrease the 75% percentile relative to GECKO. We conclude that PRESTO widens the ranges of highly constraint reactions, which causes a slight decrease in feasible range for reactions with wide feasible ranges.

In line 43, the authors mention that *in vivo* turnover numbers for *S. cerevisiae* and *A. thaliana* do not reflect *in vitro* measurements. Is this statement true for other organisms? To justify the proposed solution for turnover number correction, it would be better to clarify if the turnover number needs correction for other organisms too.

We thank the reviewer for bringing up this important point. Since estimation of *in vivo* turnover numbers requires availability of large proteomics data sets, data currently do not allow to draw a general conclusion. But all studies in eukaryotic organisms point towards inconsistencies between *in vitro* measurements and *in vivo* estimates. Most recently a study in *Chlamydomonas reinhardtii* (Arend et al. (2022), bioRxiv, <https://doi.org/10.1101/2022.11.06.515318>), has come to the same conclusion.

Can PRESTO be used to correct turnover numbers for other organisms too? If yes, then what are the parameters that need to be taken into consideration while using PRESTO for other organisms like homo sapiens, Mus musculus, etc.?

PRESTO can certainly be applied to models of other organisms. The required data are (1) protein abundances, (2) total protein contents, (3) medium definition (not required, but good to have), (4) specific growth rates, (5) (enzyme-constrained) genome-scale metabolic model. We note that the organism must be included in the KEGG database to enable the generation of an enzyme-constrained metabolic model using GECKO.

The source code provided in the github repository is not able to replicate the results in the paper without error. For instance, while running `correct_kcats_yeast.m`, there was an error at line 112. (Error using save. Variable 'deltaSamplingMat' not found).

We apologize for the mistake. The line that calls the sampling function was removed to reduce running time and had not been added again. This omission is now fixed.

Minor

Please check the typo in line 392 (returning the minimum)

We updated the manuscript accordingly.

The notation for relative error in lines 436 and 414 are e_r and ω respectively. Can you please clarify what these notations mean and how they are different?

Both variables denote the relative error. We thank the reviewer for spotting this mistake. We renamed e_r into ω to keep the notation consistent.

Reviewers' Comments:

Reviewer #1:

Remarks to the Author:

The authors have fixed most of the issues and the current manuscript is now much better than the previous version.

While the authors responded well to comment 1.1 by arguing that "PRESTO estimates corrections for k_{cat} values, not apparent catalytic rates", I do not see that they revised the manuscript accordingly. Please clarify in the manuscript that this study is just to correct the in vitro turnover numbers. A lot of passages should be rephased, e.g., "in vivo estimation of turnover number" should be changed to "correction of in vitro turnover number" (line 20), "and estimates of in vivo turnover numbers" should be removed (line 145), "in vivo estimates of turnover numbers from" to "corrected in vitro turnover numbers by" (line 377), etc.

Moreover, I have one concern regarding the term "in vivo turnover number" or "in vivo k_{cat} " used in the manuscript, which is very confusing.

Below I list several related terms:

1. "in vitro turnover number (or in vitro k_{cat})" represents turnover number by in vitro measurement.
2. "in vivo/apparent catalytic rate (or k_{app})" represents the condition-specific estimate by the pFBA method.
3. "maximum in vivo catalytic rate" represents the maximum across all conditions.
4. "in vivo turnover number (or in vivo k_{cat})" was introduced in the paper (<https://doi.org/10.1073/pnas.2001562117>), which represents the estimate by the pFBA method. In the manuscript, "in vivo turnover number (or in vivo k_{cat})" can represent the estimate by the pFBA method (e.g., line 17 and 76) but also represent the corrected in vitro turnover number by PRESTO (e.g., line 145 and 286).

To address this, I would recommend that the authors refer the estimate by the pFBA method to as in vivo turnover number and the estimate by PRESTO to as corrected in vitro turnover number. This is reasonable as the pFBA method recalculates the whole set of turnover numbers but PRESTO corrects a subset of in vitro turnover numbers.

Altogether the text containing "in vivo" should be carefully examined and revised.

Reviewer #2:

Remarks to the Author:

The authors adequately addressed the concerns on my previous revision.

Reviewer #3:

Remarks to the Author:

I appreciate the effort from the authors to clarify my doubts and address all the issues raised. I recommend the publication of the manuscript after clarifying one minor issue. Please confirm the value of ϵ . In line 486/487, the authors set the value of ϵ to $\{10\}^5$. However, in the response, they mention $\epsilon=\{10\}^{-5}$.

**Response letter to the Reviewers' comment on
"Data integration across conditions improves turnover number estimates and metabolic
predictions"**

Reviewer #1 (Remarks to the Author):

The authors have fixed most of the issues and the current manuscript is now much better than the previous version.

While the authors responded well to comment 1.1 by arguing that "PRESTO estimates corrections for k_{cat} values, not apparent catalytic rates", I do not see that they revised the manuscript accordingly. Please clarify in the manuscript that this study is just to correct the *in vitro* turnover numbers. A lot of passages should be rephased, e.g., "*in vivo* estimation of turnover number" should be changed to "correction of *in vitro* turnover number" (line 20), "and estimates of *in vivo* turnover numbers" should be removed (line 145), "*in vivo* estimates of turnover numbers from" to "corrected *in vitro* turnover numbers by" (line 377), etc.

Moreover, I have one concern regarding the term "*in vivo* turnover number" or "*in vivo* k_{cat} " used in the manuscript, which is very confusing.

Below I list several related terms:

1. "*in vitro* turnover number (or *in vitro* k_{cat})" represents turnover number by *in vitro* measurement.
2. "*in vivo*/apparent catalytic rate (or k_{app})" represents the condition-specific estimate by the pFBA method.
3. "maximum *in vivo* catalytic rate" represents the maximum across all conditions.
4. "*in vivo* turnover number (or *in vivo* k_{cat})" was introduced in the paper (<https://doi.org/10.1073/pnas.2001562117>), which represents the estimate by the pFBA method.

In the manuscript, "*in vivo* turnover number (or *in vivo* k_{cat})" can represent the estimate by the pFBA method (e.g., line 17 and 76) but also represent the corrected *in vitro* turnover number by PRESTO (e.g., line 145 and 286).

To address this, I would recommend that the authors refer the estimate by the pFBA method to as *in vivo* turnover number and the estimate by PRESTO to as corrected *in vitro* turnover number. This is reasonable as the pFBA method recalculates the whole set of turnover numbers but PRESTO corrects a subset of *in vitro* turnover numbers.

Altogether the text containing "*in vivo*" should be carefully examined and revised.

We thank the reviewer again for the thorough reading of the manuscript in order to pinpoint these ambiguities. We followed the suggestion of the reviewer and now consequently use the suggested terms to refer to certain sets of (maximum) catalytic rate constants:

- (1) "*in vitro* turnover number" refers to rate constants obtained from *in vitro* experiments,**
- (2) "apparent catalytic rate" refers to a condition-specific rate constant obtained from pFBA,**
- (3) "(estimates of) *in vivo* turnover number" refers to the maximum of these apparent catalytic rates,**

(4) “corrected turnover number” refers to the rate constants obtained from the PRESTO approach. We choose this experimental system agnostic term, since PRESTO can be used to correct any set of previous turnover numbers (*in vitro* or *in vivo*) and always relies on proteomics data from experiments.

Reviewer #2 (Remarks to the Author):

The authors adequately addressed the concerns on my previous revision.

Reviewer #3 (Remarks to the Author):

I appreciate the effort from the authors to clarify my doubts and address all the issues raised. I recommend the publication of the manuscript after clarifying one minor issue. Please confirm the value of ε . In line 486/487, the authors set the value of ε to $\{10\}^5$. However, in the response, they mention $\varepsilon=\{10\}^{-5}$.

We thank the reviewer for spotting this discrepancy. The value for $\varepsilon=10^5$ given in the manuscript text is indeed correct.

Reviewers' Comments:

Reviewer #1:

Remarks to the Author:

The issues have been addressed in the current manuscript. I do not have any comments.